# Piezo2 expressing nociceptors mediate mechanical sensitization in experimental osteoarthritis

Alia M. Obeidat[1,4], Matthew J. Wood [1,4], Natalie S. Adamczyk[1], Shingo Ishihara[1], Jun Li[1], Lai Wang[1], Dongjun Ren[2], David A. Bennett[3], Richard J. Miller[2], Anne-Marie Malfait [1] & Rachel E. Miller [1] ✉

Non-opioid targets are needed for addressing osteoarthritis pain, which is mechanical in nature and associated with daily activities such as walking and climbing stairs. Piezo2 has been implicated in the development of mechanical pain, but the mechanisms by which this occurs remain poorly understood, including the role of nociceptors. Here we show that nociceptor-specific *Piezo2* conditional knock-out mice were protected from mechanical sensitization associated with inflammatory joint pain in female mice, joint pain associated with osteoarthritis in male mice, as well as both knee swelling and joint pain associated with repeated intra-articular injection of nerve growth factor in male mice. Single cell RNA sequencing of mouse lumbar dorsal root ganglia and in situ hybridization of mouse and human lumbar dorsal root ganglia revealed that a subset of nociceptors co-express *Piezo2* and *Ntrk1* (the gene that encodes the nerve growth factor receptor TrkA). These results suggest that nerve growth factor-mediated sensitization of joint nociceptors, which is critical for osteoarthritic pain, is also dependent on Piezo2, and targeting Piezo2 may represent a therapeutic option for osteoarthritis pain control.

Nociceptors are sensory neurons that enable the body to detect noxious chemical, mechanical and thermal stimuli to avoid tissue damage. Under pathologic conditions, such as during inflammation, nociceptors become sensitized so that they respond to normally non-noxious stimuli. The concept of nociceptor sensitization has been extensively described, but the precise molecular underpinnings of sensitization to mechanical stimuli are incompletely understood. Recent work suggests that the mechanosensitive ion channel, Piezo2, contributes to a range of fundamental sensory neuron functions, including sensing of light touch[1], proprioception[2], vibratory detection[3], and more recently, mechanical pain[3,4]. In particular, it was demonstrated that in models of acute inflammation or nerve injury, *Piezo2* knock-out mice showed reduced mechanical allodynia[4]. Furthermore, individuals with PIEZO2 loss of function mutations were

unable to detect light touch applied to skin after inflammation was induced with capsaicin[3].

Although it seems increasingly clear that Piezo2 may mediate mechanical sensitization induced by inflammation in mouse models, it is not clear whether this phenomenon contributes to pain associated with specific diseases in the human population. Osteoarthritis, one of the world's most common diseases, is characterized by mechanically driven pain in the presence of tissue injury and low-grade inflammation[5–7]. Clinical research has demonstrated that people afflicted with osteoarthritis develop mechanical sensitization not only at the affected joint but also at sites away from the joint, suggesting both peripheral and central mechanical sensitization[8]. Joint replacement surgery often alleviates this sensitization and pain, suggesting that the osteoarthritic joint drives these processes, even at late stages

[1]Department of Internal Medicine, Division of Rheumatology, Rush University Medical Center, Chicago, USA. [2]Department of Pharmacology, Northwestern University, Chicago, USA. [3]Rush Alzheimer's Disease Center and Department of Neurological Sciences, Rush University Medical Center, Chicago, USA. [4]These authors contributed equally: Alia M. Obeidat, Matthew J. Wood. ✉e-mail: Rachel_Miller@rush.edu

of disease[8]. Interestingly, it has also been shown that individuals with lowered pain pressure thresholds to mechanical pressure applied at the knee are more likely to go on to develop persistent knee pain, suggesting that mechanical sensitization plays a key role in the early stages of the development of osteoarthritis pain[9].

Nerve growth factor (NGF) has been suggested as a therapeutic target for osteoarthritis pain, and clinical trials with antibodies that neutralize NGF reported positive results in terms of pain relief[10,11]. Consistent with this data, several rodent models of joint pain, including models of osteoarthritis pain, have been shown to be dependent on NGF signaling[12-15]. Additionally, using rodent models of osteoarthritis, we and others have shown that a unique subset of nociceptors ('silent nociceptors') become responsive to mechanical stimuli (indicating peripheral sensitization) during the course of disease[16-19]. In vitro, transfection of silent nociceptors with *Piezo2* targeting siRNA was sufficient to block NGF-induced mechanical sensitization of these cells[20]. Previous publications exploring the role of Piezo2 in nerve injury or acute inflammatory pain have used mice in which the channel was deleted from all sensory neurons, and therefore behavioral measures of mechanical allodynia were difficult to assess, since these mice cannot move normally due to proprioception deficits[2]. Here, we depleted *Piezo2* specifically from nociceptors by using Na$_V$1.8-Cre mice, and used these mice to examine the role of Piezo2 in the development of mechanical sensitization in experimental osteoarthritis. Our results suggest that nociceptor expression of Piezo2 plays a key role in this process and cooperates with NGF-mediated signaling in producing nociceptor sensitization. Hence, Piezo2 may represent a therapeutic target for osteoarthritis pain.

## Results

### *Piezo2* can be specifically depleted from nociceptors in mice by using the marker *Na$_V$1.8*

While recent studies suggest that Piezo2 mediates mechanical allodynia in models of acute inflammation and nerve injury[3,4], the role of Piezo2 in mediating mechanical pain associated with specific diseases in animals and humans has yet to be investigated. A major obstacle to such studies has been that the pan-sensory neuron *Piezo2* knock-out mice that have been used to date also show pronounced proprioceptive deficits, making it difficult to accurately assess evoked responses to mechanical stimuli[2]. Indeed, *Piezo2* mRNA is expressed by a wide range of sensory neurons, as seen in single cell RNA sequencing (scRNAseq) analyses of mouse and human DRGs[21,22]. We used scRNAseq to confirm that *Piezo2* is expressed by many subsets of nociceptors (Fig. 1a, b; Supplementary Fig. 1) as well as by large diameter neurons, which includes the proprioceptor subset (Fig. 1a, b Supplementary Fig. 1). Additionally, RNA-scope analysis revealed a large degree of overlap between nociceptors, defined as *Na$_V$1.8*+ neurons and *Piezo2*, with 38 ± 2% of DRG neurons expressing both *Na$_V$1.8* and *Piezo2* (Fig. 1c−e).

Therefore, we decided to create nociceptor-specific conditional *Piezo2* knock-out mice, enabling us to study the functional implications of Piezo2 expression by these neurons in particular. We crossed *Na$_V$1.8*-Cre mice[23] with *Piezo2*-loxp mice[2], in which exons 43-45 are flanked by loxp sequences. We have used both heterozygous CKO mice (*Piezo2*$^{CKOfl/+}$) and homozygous CKO mice (*Piezo2*$^{CKOfl/fl}$) for experiments in this study. RNAscope confirmed the effectiveness of this strategy; co-expression of *Na$_V$1.8* and *Piezo2* was reduced to 6 ± 1% of DRG neurons in homozygous *Piezo2*$^{CKOfl/fl}$ mice (Fig. 1c−e and

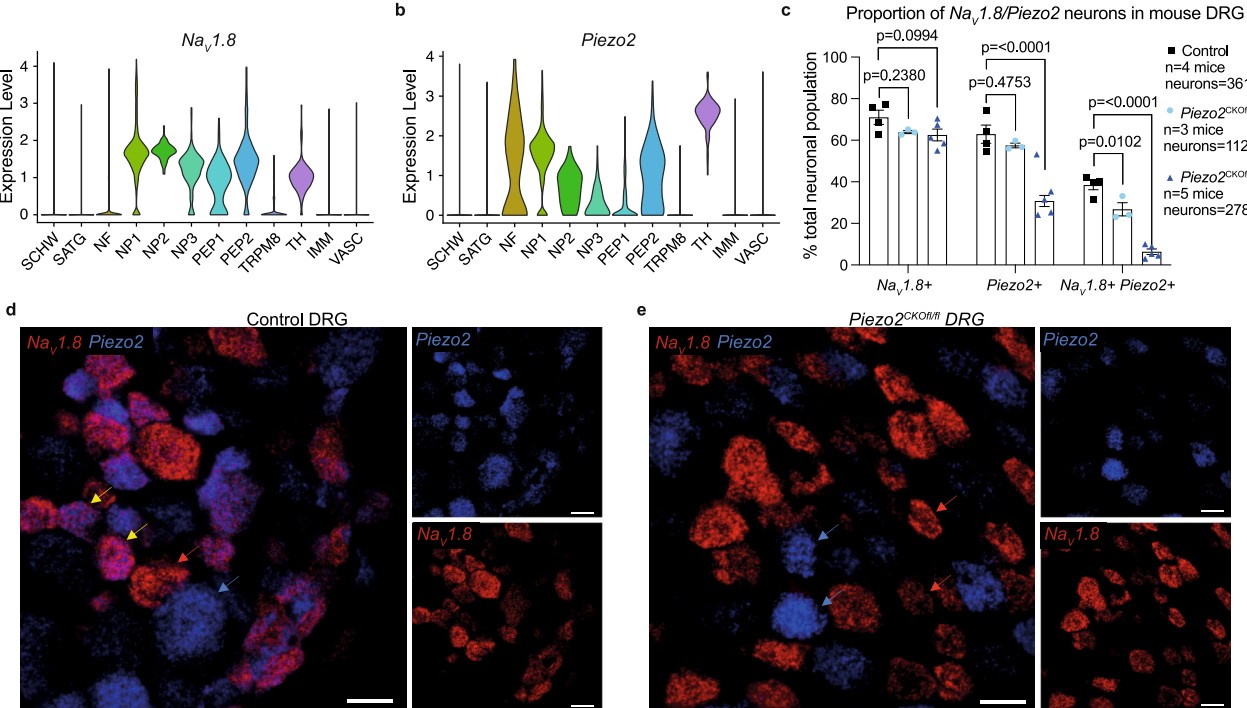

**Fig. 1 | Piezo2 is depleted from nociceptors using the marker Na$_V$1.8. a, b** ScRNA-seq of naive L3-L5 dorsal root ganglion (DRG) cells demonstrates that *Piezo2* is expressed by a subset of nociceptors (*Na$_V$1.8* + neurons) in addition to large diameter neurons (NF; *Na$_V$1.8*− neurons); SCHW Schwann cells, SATG satellite glia, NF neurofilament, NP non-peptidergic nociceptors, PEP peptidergic nociceptors, TRPM8 transient receptor potential melastatin 8, TH tyrosine hydroxylase containing, IMM immune cells, VASC vascular cells. **c−e** *Piezo2* was deleted in nociceptors by using the marker *Na$_V$1.8*. RNAscope was used to quantify the number of cells expressing *Na$_V$1.8* and *Piezo2*. There is reduced co-expression of *Na$_V$1.8* + / *Piezo2* + in heterozygous *Piezo2*$^{CKOfl/+}$ (cyan circles) and homozygous *Piezo2*$^{CKOfl/fl}$

(blue triangles) mice compared to control mice (black squares) (greater than 12 weeks of age). **c** One-way ANOVA with Dunnet's multiple comparison test was used to compare numbers of cells between strains. Mean ± SEM. **d** Representative sections from control and **e** *Piezo2*$^{CKOfl/fl}$ mice. Scale bar = 25 μm. Yellow arrow indicates example cells expressing *Na$_V$1.8* and *Piezo2*. Red arrow indicates example cells only expressing *Na$_V$1.8*. Blue arrow indicates example cells expressing only *Piezo2*. n = 4 no Cre controls; n = 3 *Piezo2*$^{CKOfl/+}$; n = 5 *Piezo2*$^{CKOfl/fl}$. Adjustments to individual colour channels on merged images was performed using brightness and contrast tools. Adjustments were made to the entire image and have been applied across all images and controls. For **c**, source data are provided as a Source Data file.

Supplementary Figs. 2, 3). In heterozygous *Piezo2*CKOfl/+ mice, the number of co-expressing *Na*V*1.8* and *Piezo2* neurons was also reduced (27 ± 3%) compared to controls, but to a lesser extent than homozygous mice, as expected (Fig. 1c, and Supplementary Figs. 2, 3).

Previous work has shown that *Piezo2* inhibition alters C and Aδ nociceptor responses to mechanical stimuli ex vivo[4] and in select subclasses of nociceptors in vivo[24,25]. Ex vivo, electrophysiology using a skin-nerve preparation demonstrated that both C and Aδ fibers had reduced firing in response to mechanical stimuli in *Piezo2* deficient mice[4]. In vivo, partial knockdown of Piezo2 in rat DRG via intrathecal injection of Piezo2 antisense oligodeoxynucleotides reduced Aδ fiber discharge frequency of bone afferents in response to pressure applied to the marrow cavity[24]. Likewise, AAV mediated depletion of Piezo2 in mice reduced responses to a brush applied to the cheek as assessed by in vivo calcium imaging of the trigeminal ganglia[25]. Recent work has also shown that heterozygous introduction of a gain of function Piezo2 mutation into DRG neurons is sufficient to increase DRG mechanosensitivity in vivo[26]. To confirm these observations, we used GCaMP6s expressing lines of the newly generated heterozygous *Piezo2*CKOfl/+ mice (NaV1.8-Cre+/-; GCaMP6s loxpfl/+; Piezo2 loxpfl/+ mice) and performed in vivo calcium imaging of the L4 DRG to examine nociceptor responses to 30 or 100 g of mechanical force applied to the knee of anesthetized mice (Fig. 2a). By imaging the L4 DRG we were able to observe the responses of a large population of sensory neurons that innervate the knee joint[18,27]. We applied two different levels of mechanical force (30 g and 100 g) to the knee joints, and observed reduced numbers of neurons responding to these forces in naive *Piezo2*CKOfl/+ mice (Fig. 2b–f; Supplementary movies 1, 2). In addition, the size of the responses quantified by peak area under the curve was reduced with the 100 g force in the *Piezo2*CKOfl/+ mice (Fig. 2g). In both the control and *Piezo2*CKOfl/+ mice, the responding neurons were primarily in small- to medium-diameter neurons (Fig. 2h), however, the neurons that still responded in the *Piezo2*CKOfl/+ mice were larger than those of the controls. Finally, as an additional control, we also applied a dynamic brush stimulus to the hindpaws of these mice to target the C-low threshold mechanoreceptor population. Similar to what has been reported when brushing the cheek[3], these responses were in small diameter neurons in control mice, and responses to this stimulus were reduced in *Piezo2*CKOfl/+ mice (Supplementary Fig. 4). Collectively, these data suggest that the *Piezo2*CKOfl/+ and *Piezo2*CKOfl/fl mice are suitable for studying the role of Piezo2 specifically expressed by nociceptors in mouse models of painful conditions.

## Depletion of *Piezo2* from nociceptors protects mice from mechanical sensitization associated with acute joint inflammation

Prior in vitro work has demonstrated that Piezo2 activity can be enhanced on short time scales by inflammatory molecules such as bradykinin[28,29]. Therefore, we decided to test the role of Piezo2 in mediating pain associated with acute inflammation in the knee joint following a single injection of complete Freund's adjuvant (CFA). As others have demonstrated[30], a single injection of CFA caused rapid knee swelling in female wild-type mice (Fig. 3a, b). Accompanying this swelling, wild-type mice developed knee hyperalgesia (Fig. 3c, d), which resolved as the swelling resolved through 21 days after the injection. Female homozygous *Piezo2*CKOfl/fl mice injected with CFA also developed rapid knee swelling (Fig. 3a, b), but they developed less knee hyperalgesia compared to wild-type mice (Fig. 3c, d), suggesting that Piezo2 may play a role in vivo in mediating mechanical sensitization in response to acute joint inflammation.

## Depletion of *Piezo2* from nociceptors protects mice from osteoarthritis mechanical sensitization

Osteoarthritis pain is highly mechanical in nature, and sensitization to mechanical stimuli is a key feature of experimental osteoarthritis models as well as the human disease[31]. In addition, in contrast to the previous experiment that modeled response to acute joint inflammation, osteoarthritis is a disease that develops over decades and is associated with chronic, low levels of inflammation. Therefore, we decided to investigate if Piezo2 contributes to the development of mechanical sensitization associated with two validated models of experimental osteoarthritis (destabilization of the medial meniscus (DMM) surgery and spontaneous osteoarthritis associated with aging). We and others have previously characterized pain-related behaviors associated with mechanical input over the course of these slowly developing models of osteoarthritis. Mice show a reduced threshold for mechanically induced pain behaviors after DMM surgery, including hyperalgesia at the injured knee and mechanical allodynia at the ipsilateral hind paw[32–35]. In a pilot experiment, we performed DMM surgery in both heterozygous *Piezo2*CKOfl/+ and homozygous *Piezo2*CKOfl/fl mice as well as in littermates lacking *Na*V*1.8*-Cre as a control. Importantly, these mice did not exhibit signs of proprioceptive deficits due to *Piezo2* deletion (Supplementary Fig. 5a) and had a normal body weight (Supplementary Fig. 5b). Protection against mechanical allodynia of the hind paw was observed 4 and 8 weeks after DMM surgery in the homozygous *Piezo2*CKOfl/fl mice, with a trend in protection in the heterozygous *Piezo2*CKOfl/+ mice (Supplementary Fig. 5c).

To investigate this further, a second independent experiment was performed using homozygous *Piezo2*CKOfl/fl mice and NaV1.8-Cre+/- controls. Homozygous *Piezo2*CKOfl/fl mice were protected from both knee hyperalgesia and hind paw mechanical allodynia through 8 weeks after DMM (Supplementary Fig. 6a, b), suggesting that Piezo2 expressing nociceptors are important for development of mechanical sensitization in this model. In both of these experiments, cartilage damage and osteophyte formation were comparable in *Piezo2*CKOfl/fl and control mice, indicating that protection from mechanical sensitization did not result in exacerbated joint damage (Supplementary Figs. 5d–f; 6c–f).

Because osteoarthritis pain is associated with weight-bearing activities of daily living[6,36], we sought to directly assess the role of Piezo2 in generating pain associated with weight-bearing activities in rodents. Therefore, we developed a voluntarily accessed static incapacitance meter test[37] in which hind leg weight-bearing forces are measured while mice perform a simulation of a natural task in this species—grasping and pulling on fibers to build nests or obtain food. In a laboratory setting, this task has been mimicked using string to assess arm and hand movements[38]. Here, we have combined this string-pulling task with assessment of hind limb weight-bearing by training the animals to perform this task while standing on force plates. Following DMM surgery, wild-type mice began to put less weight on the operated hind limb by 12 weeks post surgery ($p = 0.0176$ vs. pre-surgery; Fig. 4a), which further developed by 16 weeks post surgery ($p = 0.0100$ vs. pre-surgery), consistent with previous data generated using a traditional static incapacitance meter[39]. In contrast, *Piezo2*CKOfl/fl did not develop any changes in weight-bearing on the operated limb through 16 weeks. Because mechanical sensitization has been associated with weight-bearing pain[40], we also assessed knee hyperalgesia concurrently in this cohort. Similar to the results shown in Supplementary Fig. 6, wild-type mice developed knee hyperalgesia by 4 weeks after DMM surgery, and this was sustained through 16 weeks (vs. pre-surgery: 4 weeks, $p < 0.0001$; 8 weeks, $p < 0.0001$; 12 weeks, $p < 0.0001$; 16 weeks, $p = 0.0041$) (Fig. 4b). In contrast, *Piezo2*CKOfl/fl mice failed to develop knee hyperalgesia through 16 weeks after DMM surgery (vs. pre-surgery: 4 weeks, $p = 0.5167$; 8 weeks, $p = 0.9514$; 12 weeks, $p = 0.7717$; 16 weeks, $p = 0.2758$).

In addition to joint injury, age is also a major independent risk factor for osteoarthritis[7], and C57BL/6 male mice have been shown to develop age-associated cartilage damage[41]. At the age of 18 months, male littermate controls (NaV1.8-Cre-/-) had a reduced hind paw withdrawal threshold to mechanical stimuli compared to homozygous

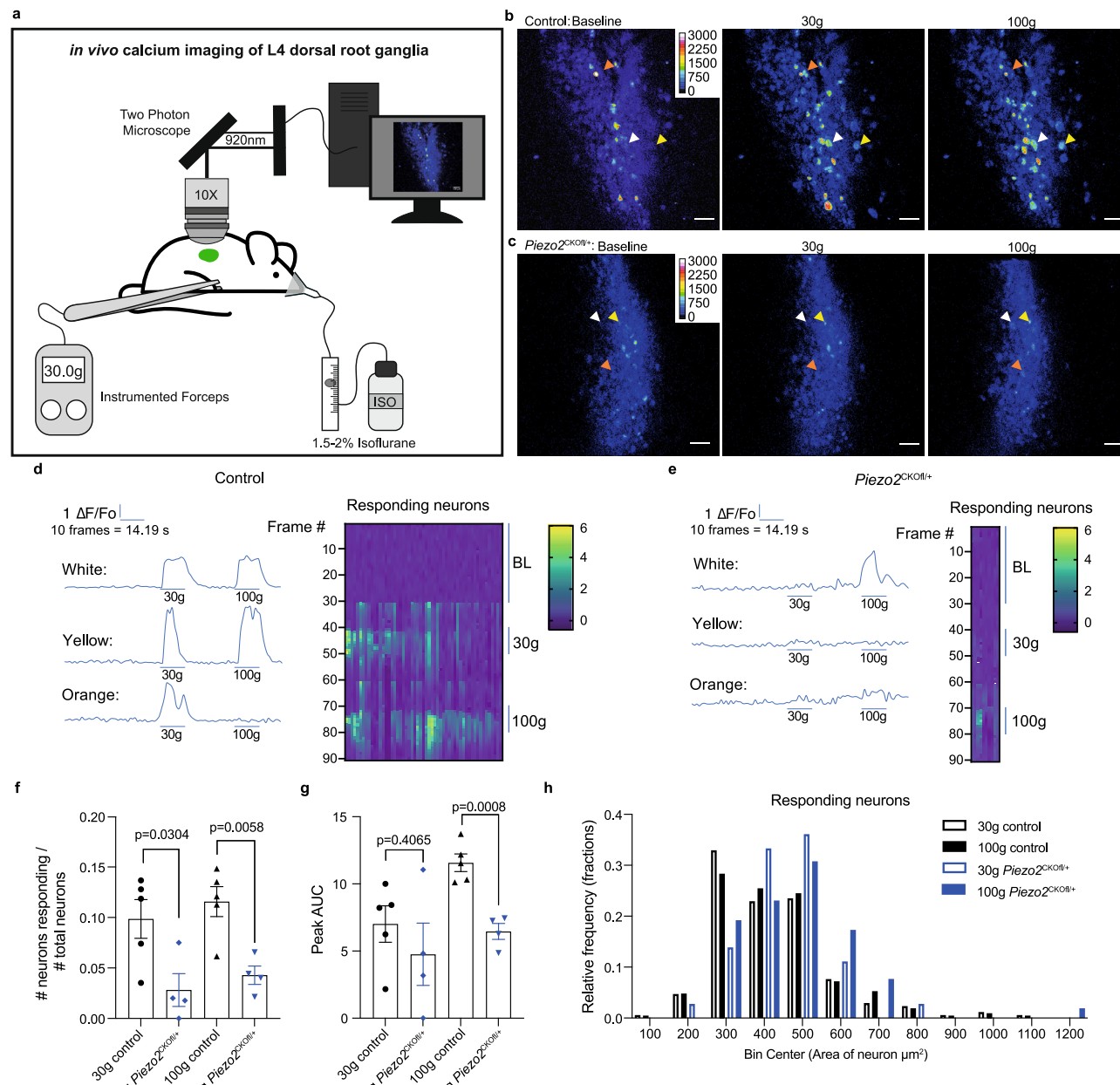

**Fig. 2 | In vivo calcium imaging demonstrates a role for nociceptor expression of Piezo2 in mediating intracellular calcium responses to mechanical force applied to the knee joint. a** The L4 dorsal root ganglion (DRG) is imaged by two photon microscopy in anesthetized mice while mechanical stimuli are applied to the knee joint using an instrumented forceps. **b**, **c** Representative DRG images and **d**, **e** individual neuronal traces from control ($Na_V1.8;GCaMP6s$) and $Piezo2^{CKOfl/+}$ ($Na_V1.8;GCaMP6s;Piezo2^{fl/+}$) mice. Corresponds to Supplementary Movies 1 and 2. Heatmaps depict changes in fluorescence for responding neurons (each column is one responding neuron; each row is one frame). **f** The number of $Na_V1.8+$ neurons

responding to each stimulus was quantified and compared between strains by unpaired two-tailed $t$-test (each dot = one mouse; for each mouse >225 neurons imaged in total) (control, $n = 5$ mice (black; 30 g = circles; 100 g = triangles); $Piezo2^{CKOfl/+}$, $n = 4$ mice (blue; 30 g = diamonds; 100 g = nabla)). Mean±SEM. **g** The peak area under the curve (AUC) of each neuron responding to each stimulus was quantified, averaged for each mouse, and compared among strains by unpaired two-tailed $t$-test. (same mice as in **f**: control, $n = 5$ mice; $Piezo2^{CKOfl/+}$, $n = 4$ mice). Mean ± SEM. **h** The relative frequency distribution of the areas ($\mu m^2$) of responding neurons. For **f**–**h**, source data are provided as a Source Data file.

$Piezo2^{CKOfl/fl}$ mice (Fig. 4c), once again indicating the role of this channel in mechanical sensitization in osteoarthritis. Both genotypes developed similar levels of age-associated cartilage damage and osteophyte formation in the knee joints when assessed at age 22 months (Supplementary Fig. 7).

Finally, to investigate whether targeting Piezo2 may represent a suitable strategy for relieving osteoarthritis pain, we used chemogenetics to acutely silence *Piezo2 +* nerves within the knee joint (heterozygous Piezo2-Cre$^{+/-}$;Pdi$^{fl/+}$ mice). With this genetic strategy, we observed an expression pattern of the inhibitory DREADD

receptor in DRG neurons consistent with the broad pattern of expression of Piezo2 by DRG neurons (Figs. 1, 5a and Supplementary Fig. 8a). Nine weeks after DMM surgery, when knee hyperalgesia was present ('pre' time point in Fig. 5b), we found that intra-articular injection of clozapine-N-oxide (CNO) transiently reversed knee hyperalgesia compared to saline (Fig. 5b). This suggests that local inhibition of *Piezo2+* neurons may represent a therapeutic route. A control experiment in the CFA model confirmed the specificity of CNO for the Pdi receptor (Supplementary Fig. 8b).

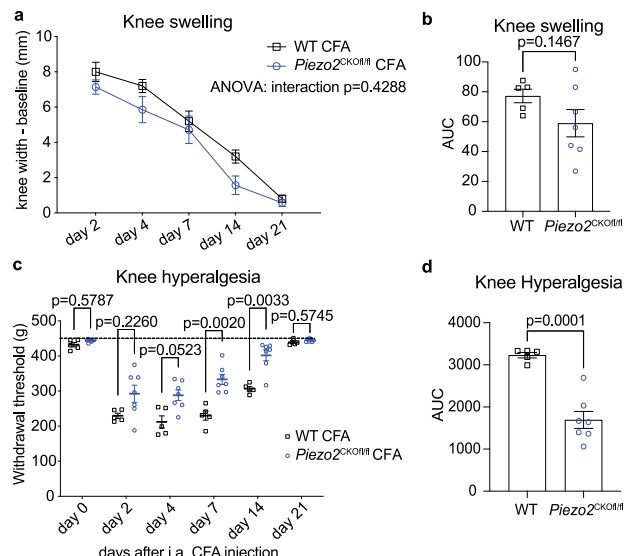

**Fig. 3 | Piezo2 depletion protects from CFA induced mechanical sensitization. a** Knee swelling was assessed in female wild-type (WT) (black squares) or homozygous *Piezo2*[CKOfl/fl] (blue circles) mice given a single intra-articular injection of Complete Freund's adjuvant (CFA) (i.a. 5 μg, 5 μL). Two-way repeated measures ANOVA. **b** Area under the curve analysis over the time course was used to assess knee swelling. Unpaired two-tailed *t*-test. **c** Knee hyperalgesia was assessed in the same mice as part **a**. Dashed line indicates maximum of the assay: 450 g. Two-way repeated measures ANOVA with Sidak's post-test. **d** Area under the curve analysis over the time course was used to assess knee hyperalgesia. Unpaired two-tailed *t*-test. Mean ± SEM. For **a**–**d** WT (*n* = 5) and homozygous female *Piezo2*[CKOfl/fl] (*n* = 7) mice. i.a. CFA intra-articular Complete Freund's adjuvant. For **a**–**d**, source data are provided as a Source Data file.

## A subset of nociceptors co-express *Piezo2* and *Ntrk1* in mouse and human DRG

Mechanical sensitization is a major feature of osteoarthritis pain[8] and may be mediated by several inflammatory factors present in the osteoarthritis joint, including cytokines and growth factors that are synthesized and released as a result of ongoing tissue damage[42]. In particular, the neurotrophin NGF, which is upregulated in osteoarthritic joint tissues[39,43,44] and has been shown to play an important role in osteoarthritis pain[43], produces rapidly sensitizing effects in nociceptors through activation of its high-affinity receptor, tropomyosin receptor kinase A (TrkA)[45]. A recent study observed that application of NGF to mouse DRG neurons in culture greatly increased their mechanosensitivity, and *Piezo2* siRNA inhibited this response[20]. Additionally, rapid mechanical hypersensitivity induced by local injection of NGF into bone marrow could be reduced by *Piezo2* knockdown in rats[24]. Hence, a precise description of the Piezo2 and TrkA signaling axis in the joint is expected to increase our understanding of how pain is produced in osteoarthritis.

RNAscope of the lumbar DRGs of adult healthy mice indicated that *Na_V_1.8*, *Piezo2* and *Ntrk1* (gene that encodes TrkA) are co-expressed by 16 ± 0.9% of all DRG neurons (Fig. 6a, b, e, Supplementary Fig. 9). Analysis by scRNA-seq also found a large amount of co-expression—by this method we found that *Piezo2* and *Ntrk1* were co-expressed in 38% of *Na_V_1.8* + DRG neurons (Supplementary Fig. 10a), including neurons in various sub-types of nociceptors (Supplementary Fig. 10b, c).

In human lumbar level DRG, a similar pattern of co-expression was observed by RNAscope. In DRGs collected *post mortem* from three donors (male *n* = 1, female *n* = 2, ages 82–94, BMI, 21.2-27.6), examination of a total of 486 neurons showed that *Na_V_1.8*, *PIEZO2* and *NTRK1* were co-expressed by 19 ± 4% of all DRG neurons (Fig. 6c–e, Supplementary Figs. 11–13). Examining the combinations of single,

double and triple positive expression patterns of these genes does reveal some differences between human and mouse. As has been noted by another study[46], *NTRK1* is more broadly expressed by *Na_V_1.8* + cells in human compared to mouse (Fig. 6f). In addition, *PIEZO2* is more commonly expressed together with *Na_V_1.8* and/or *NTRK1* in human neurons (12% single expression of *PIEZO2*) compared to mouse (27% single expression of *Piezo2*) (Fig. 6f). We also characterized the size of these different neuronal subsets in both murine (Fig. 6g) and human (Fig. 6h) DRG. In both species, single positive *PIEZO2* expressing neurons were larger diameter cells than single positive *Na_V_1.8* expressing neurons, consistent with the idea that this population of *PIEZO2*+/ *Na_V_1.8*-/*NTRK1*− neurons represents proprioceptors. In human, the *Na_V_1.8*+/*NTRK1*+ population consisted of smaller cells compared to the *Na_V_1.8*+/*PIEZO2*+ and *Na_V_1.8*+/*NTRK1*+/ *PIEZO2*+ populations, however this was less clear in mouse. Together, this suggests that this population of triple-expressing neurons may be translationally relevant.

## Depletion of Piezo2 blocks NGF-induced knee swelling and mechanical sensitization

Local injection of NGF into the hind paw or the knee joint has been shown to cause swelling as well as thermal and mechanical sensitization[47–49]. The exact mechanisms by which NGF exerts these effects are not clearly understood, and NGF can potentially act through numerous mechanisms including enhancing release of inflammatory mediators, transactivating nociceptor ion channels, and longer-term effects on gene expression[50]. NGF has also been shown to play an important role in mediating osteoarthritis pain[43]. In order to examine how NGF directly impacts mechanical sensitization in the knee joint in concert with Piezo2 over an extended time period, and whether this involves activation of Piezo2, we modeled this interaction by injecting recombinant murine NGF bi-weekly (500 ng NGF in 5 μL) into the knee joint cavity of adult naive wild-type or heterozygous *Piezo2*[CKOfl/+] (Na_V_1.8-Cre[+/−];GCaMP6s loxp[fl/+];Piezo2 loxp[fl/+]) mice over the course of 8 weeks. This intra-articular treatment protocol produced knee swelling in wild-type naive mice by day 7, and swelling was sustained through the end of the study on day 57 (Fig. 7a, b). In contrast, *Piezo2*[CKOfl/+] mice were protected from NGF-induced knee swelling at all time points (Fig. 7a, b). Wild-type mice developed knee hyperalgesia by week 2, and this continued through week 8 (Fig. 7c, d). In contrast, while *Piezo2*[CKOfl/+] mice also developed early knee hyperalgesia at the 2-week time point, these mice were protected from sustained knee hyperalgesia, consistent with their lack of knee swelling (Fig. 7c, d). Previously, mechanical and thermal sensitization induced by NGF has been linked to upregulation of neuropeptide expression in DRG neurons[51]. To explore this possible link, we performed an additional experiment, in which we used qPCR to analyze the expression of *Calca* (the gene encoding CGRP) in the L3-L5 DRG after 3 intra-articular injections of vehicle or NGF over the course of 1.5 weeks. In wild-type mice, NGF injection resulted in increased DRG levels of *Calca* compared to vehicle, whereas *Piezo2*[CKOfl/+] mice were protected from this effect (Fig. 7e). Together, these experiments support the idea that NGF induces its downstream effects in concert with Piezo2.

## Discussion

Our observations support a key role for Piezo2 expressed by nociceptors in mediating mechanical sensitization associated with a mouse model of acute inflammatory knee pain, two mouse models of osteoarthritis, as well as with a model induced by local injection of NGF into the knee over 8 weeks. Consistent with this interaction, co-expression of *Piezo2* and *Ntrk1* was demonstrated in subsets of murine as well as human nociceptors. These results support previous work suggesting that Piezo2 is expressed by nociceptors[4,20,24,25,52]. However, its primary function on nociceptors does not appear to be the sensation of high intensity mechanical forces under healthy conditions;

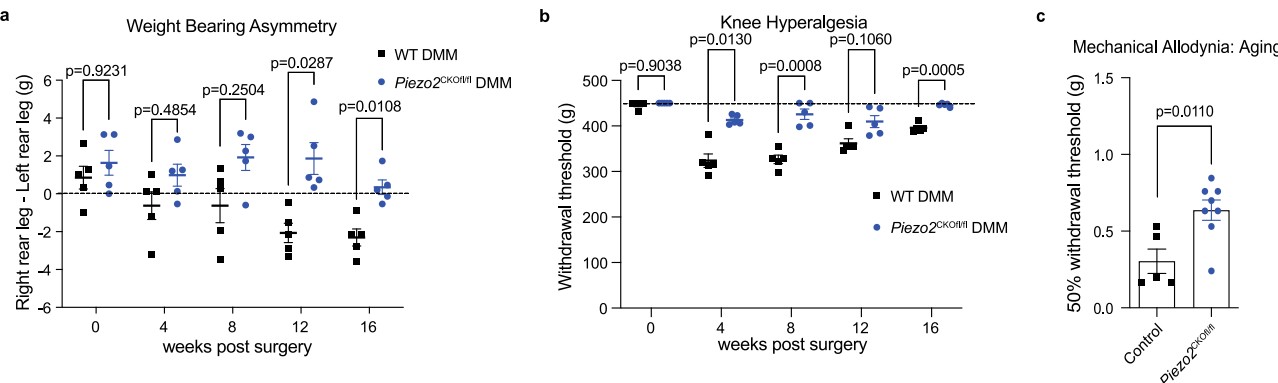

**Fig. 4 | Piezo2 plays a role in mechanical sensitization in two mouse models of osteoarthritis. a** Weight-bearing asymmetry was assessed in wild-type (WT) (*n* = 5; black squares) and *Piezo2*^CKOfl/fl (*n* = 5; blue circles) male mice after DMM surgery. Dashed line indicates 0 g which would represent mice putting equal weight on both hind legs; negative numbers indicate mice are putting more weight on the uninjured left rear leg. Two-way repeated measures ANOVA with Sidak post-test. Mean ± SEM. DMM destabilization of medial meniscus. **b** Knee hyperalgesia was assessed in mice from part **a**. Dashed line indicates maximum of the assay: 450 g. Ipsilateral operated right leg data is plotted. Contralateral unoperated left

legs: 4 weeks (mean ± SEM): wild-type (425 ± 2) and *Piezo2*^CKOfl/fl (437 ± 7). 8 weeks: wild-type (439 ± 2) and *Piezo2*^CKOfl/fl (445 ± 5). 12 weeks: wild-type (437 ± 2) and *Piezo2*^CKOfl/fl (432 ± 7). 16 weeks: wild-type (441 ± 2) and *Piezo2*^CKOfl/fl (446 ± 4). Two-way repeated measures ANOVA with Sidak post-test. An independent experiment is shown in Supplementary Fig. 6. **c** Hind paw mechanical allodynia was assessed in naive control mice (littermate no cre, *n* = 5, black squares) or *Piezo2*^CKOfl/fl (*n* = 8, blue circles) at 1.5 years of age. Histology shown in Supplementary Fig. 7. Unpaired two-tailed *t*-test on log-transformed data. Mean ± SEM. For **a–c**, source data are provided as a Source Data file.

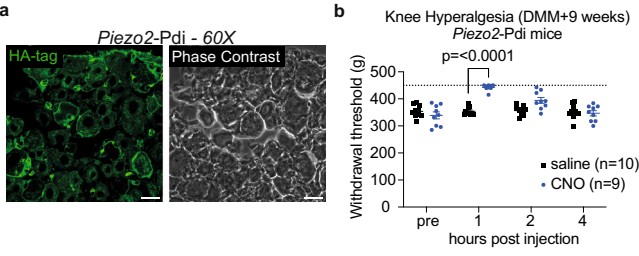

**Fig. 5 | Silencing Piezo2+ neurons transiently reverses knee hyperalgesia. a** Piezo2-Cre^+/−;Pdi^fl/+ mice express the inhibitory DREADD receptor on dorsal root ganglia (DRG) neurons. Representative images showing immunofluorescence for the HA-tagged DREADD receptor. Scale bar = 25 μm. Piezo2-Pdi mice (*n* = 6, 22–31 weeks old). **b** Knee hyperalgesia in male Piezo2-Cre^+/−;Pdi^fl/+ inhibitory DREADD mice 9 weeks after DMM surgery was assessed before, 1, 2, and 4 h after intra-articular injection of saline (*n* = 10 mice; black squares) or CNO (*n* = 9 mice; blue circles). Dashed line indicates maximum of the assay: 450 g. Two-way repeated measures ANOVA with Sidak post-test. Mean ± SEM. DMM destabilization of medial meniscus, CNO Clozapine-N-oxide. HA hemagglutinin. 'Designer Receptors Exclusively Activated by Designer Drugs' = DREADD. For **b**, source data are provided as a Source Data file.

rather Piezo2 appears to become sensitized in settings of inflammation and tissue damage. This supports previous work demonstrating that pan-sensory neuron deletion of *Piezo2* reduced mechanical sensitization following acute application of capsaicin and in a model of nerve injury[4]. In addition, individuals with *PIEZO2* loss of function mutations had reduced mechanical allodynia following capsaicin application to the skin[3]. Osteoarthritis is associated with mechanical sensitization and joint pain with movement. Joint damage products released as a result of ongoing tissue remodeling in osteoarthritis, including NGF, have been implicated in the development of mechanical sensitization[31,42], and NGF signaling has been shown to play an important role in generating osteoarthritis pain[43]. Our findings suggest that Piezo2 is an essential component of this phenomenon.

Previous work has shown that *Piezo2* depletion alters C and Aδ nociceptor responses to mechanical stimuli ex vivo[4] and in select subclasses of nociceptors in vivo[24,25]. In addition, depletion of *Piezo2* from TRPV1-lineage neurons reduced the number of action potentials fired when the colon was circumferentially stretched[53]. Here, by using

in vivo calcium imaging of the DRG, we also observed that nociceptor responses to 30 or 100 g of mechanical force applied to the knee joint were decreased in anesthetized *Piezo2*^CKOfl/+ mice. However, we found that nociceptor deletion of Piezo2 only impacts behavioral responses of mice after joint damage or inflammation has been initiated. This is consistent with other work suggesting that in the absence of inflammation, high intensity mechanical stimuli are detected by an as yet unidentified channel/receptor[25,54,55].

Other work has shown that Piezo2 activity can be enhanced on short time scales by inflammatory molecules such as bradykinin[28,29] and NGF[20,24], but exactly which signaling pathways are involved remains unclear[56–58]. Here we have shown that this is also relevant in vivo by demonstrating that *Piezo2*^CKOfl/fl mice have reduced knee hyperalgesia induced by injecting CFA into the knee joint. On short time scales, NGF has been shown to trans-activate other channels such as TRPV1. However, chronic sensitization, as assessed in this study through models of experimental osteoarthritis or through repeated intra-articular injections of NGF over the course of 8 weeks, likely involves retrograde transport of NGF to the DRG, where it can promote changes in gene expression and/or membrane localization of channels and upregulation of neuropeptides. Previous work demonstrated that NGF-induced edema was in part generated through neuropeptide release by capsaicin-sensitive nociceptors[47]. As in the current study, a mechanical component appeared to be involved in this effect since rats that were anesthetized had less edema than awake animals[47]. We also found that *Piezo2* deletion inhibits NGF-induced DRG gene expression of *Calca*, the gene encoding the neuropeptide CGRP. Hence, our results suggest that chronic NGF-mediated sensitization of joint nociceptors, which is critical for osteoarthritic pain, is also dependent on Piezo2.

While nociceptor expression of Piezo2 appears to play an important role in mechanical sensitization in osteoarthritis, inhibition of this pathway did not impact the development of cartilage degeneration, osteophytes, or synovitis after DMM surgery, nor did it prevent age-associated osteoarthritis. Many other cell types in the joint play an important role in the development of joint structural damage. In contrast to Piezo2, Piezo1 is much more broadly expressed by non-neuronal cells in mice, including by cells such as chondrocytes[59–61] and osteoblasts[62]. Constitutive deletion of *Piezo1* from chondrocytes using *Col2a1*-Cre mice impacted trabecular bone formation[60], but the effect of Piezo1 inhibition in adult animals with

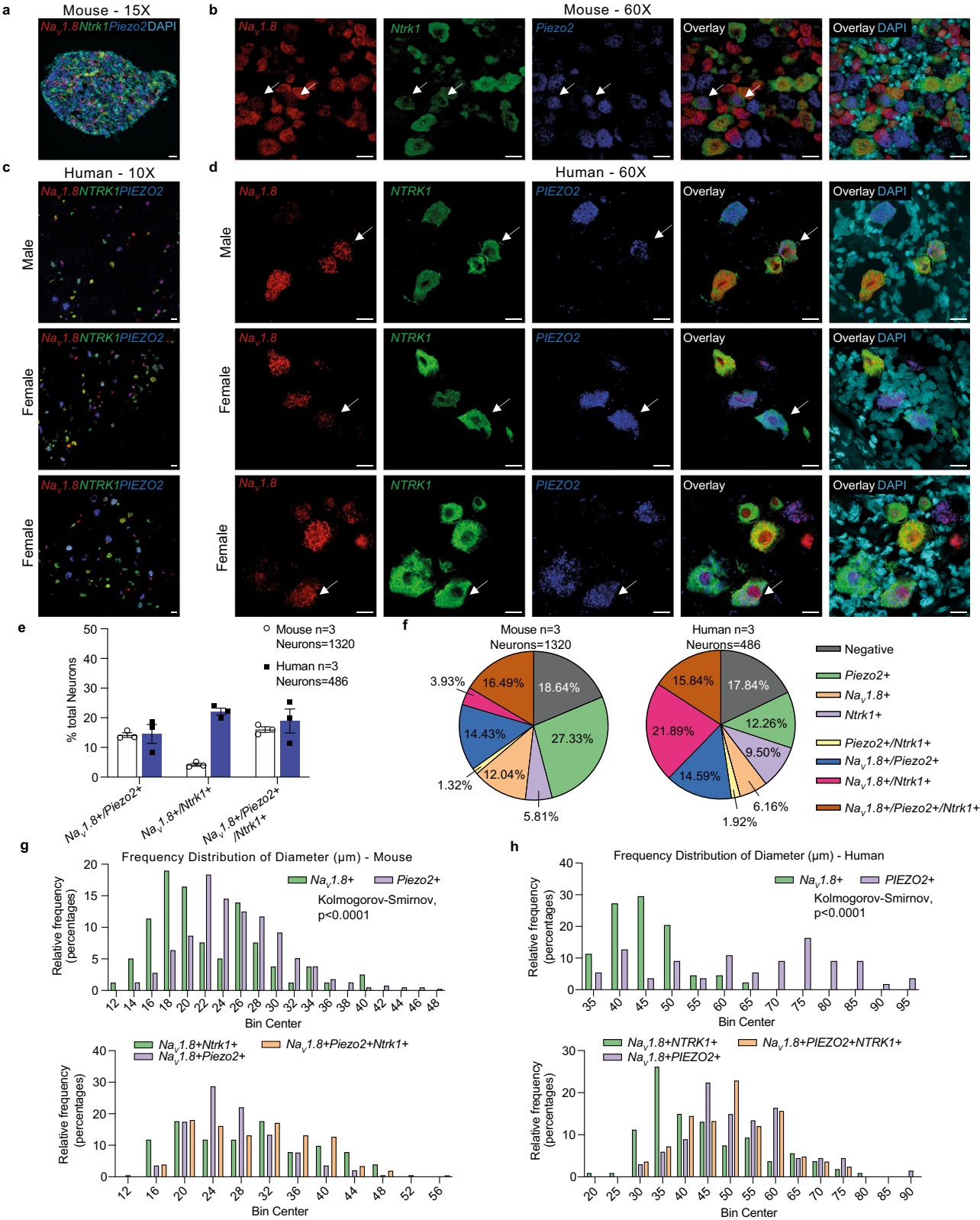

respect to development of osteoarthritis structural damage has yet to be tested.

Our study has some limitations. Since we targeted nociceptors broadly by using Na$_V$1.8-Cre mice, we did not functionally assess the impact of Piezo2 expressed by different subtypes of nociceptors. However, our transcriptional analyses help to inform the most likely subsets contributing to the observed phenotype. Both the 'NP1' and

'PEP2' subclasses of nociceptors had strong co-expression of *Scn10a* and *Piezo2*, while only the 'PEP2' type had strong co-expression with *Ntrk1* as well. In addition, recent comparative analyses of subsets of nociceptors in humans vs. mice have revealed that there are marked species differences in the exact subsets of nociceptors[22,46,63]. While we show here using RNAscope that subsets of human DRG neurons do co-express *Na$_V$1.8*, *PIEZO2*, and *NTRK1*, more work must be done to

**Fig. 6 | A subset of sensory neurons co-expresses *Na*$_V$*1.8, Piezo2,* and *Ntrk1* in both mouse and human dorsal root ganglia (DRG).** RNAscope used to identify cells expressing *Na*$_V$*1.8, Ntrk1* and *Piezo2* with DAPI in **a, b** wild-type mouse DRG (*n* = 3 mice; two male and one female) or **c, d** *Na*$_V$*1.8, NTRK1* and *PIEZO2* with DAPI in human DRG (*n* = 3 donors; one male and two females); **a, b** Representative sections of mouse DRG. White arrows indicate example cells expressing *Na*$_V$*1.8, Ntrk1* and *Piezo2*. **c, d** Representative sections of human DRG. White arrows indicate example cells expressing *Na*$_V$*1.8, NTRK1* and *PIEZO2*. Scale bar = 50 μm (**a, c**) and 25 μm (**b, d**). **e, f** Number of cells expressing *Na*$_V$*1.8, Ntrk1* and *Piezo2* in mouse DRG (white

circles) or *Na*$_V$*1.8, NTRK1* and *PIEZO2* in human DRG (black squares) determined by RNAscope. Mean ± SEM (**e**); percentages of whole (**f**). **g** Relative frequency distribution of cell subset diameters in mouse and **h** human DRG. Distributions assessed by two-tailed Kolmogorov–Smirnov test. For **a–d**, adjustments to individual colour channels on merged images was performed using brightness and contrast tools. Adjustments were applied to the entire image and have been applied across all images and controls. For **a, b**, *n* = 3 independent repeats, for **c, d**, *n* = 3 independent repeats. For **e–h**, source data are provided as a Source Data file.

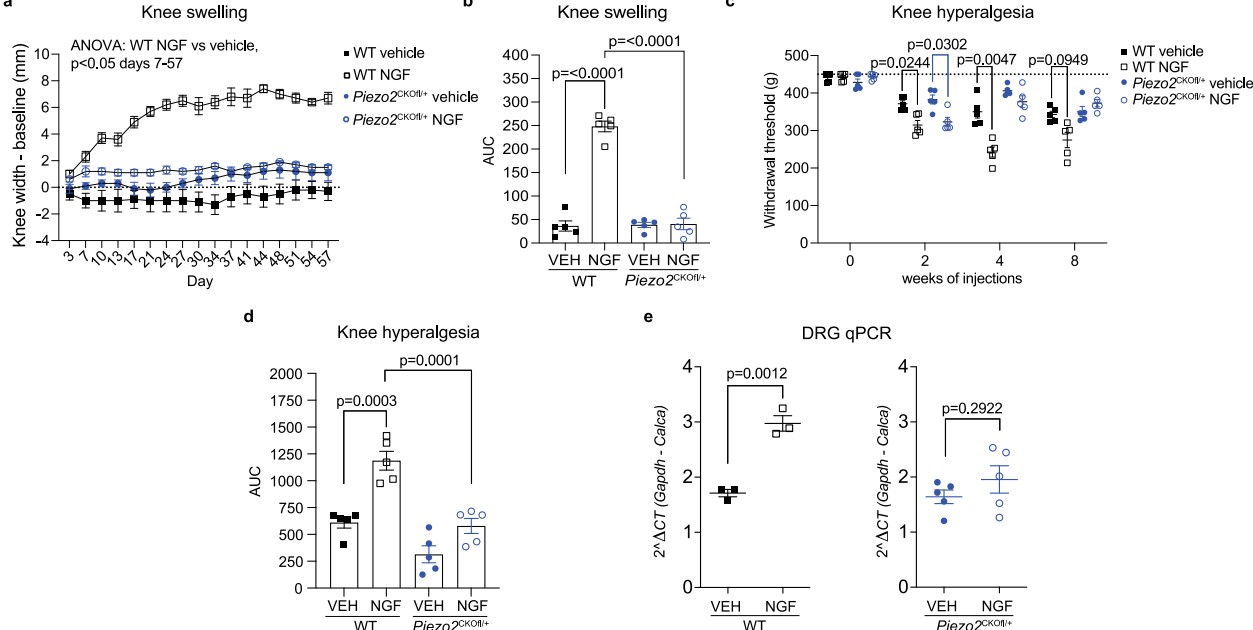

**Fig. 7 | Piezo2 depletion protects from NGF induced knee swelling and mechanical sensitization. a** Knee swelling was assessed in wild-type (WT) or heterozygous *Piezo2*$^{CKOfl/+}$ male mice given repeated intra-articular injections of recombinant murine NGF (i.a. 500 ng, 5 μL, 2x/week; WT = white squares; *Piezo2*$^{CKOfl/+}$ = white circles) or vehicle (5 μL, 2x/week; WT = black squares; *Piezo2*$^{CKOfl/+}$ = blue circles) for 8 weeks; *n* = 5 mice/group. Repeated measures two-way ANOVA with Sidak's post-test: WT NGF vs. vehicle, *p* < 0.0479 from day 7 onward. *Piezo2*$^{CKOfl/+}$, no differences between NGF vs. vehicle. **b** For the same mice as in **a**, area under the curve analysis over the time course was used to assess knee swelling. Two-way ANOVA with Sidak's post-test. (*Piezo2*$^{CKOfl/+}$

vehicle vs. *Piezo2*$^{CKOfl/+}$ NGF: *p* > 0.99). **c** Knee hyperalgesia was assessed in the same mice as part **a**. Dashed line indicates maximum of the assay: 450 g. Repeated measures two-way ANOVA with Sidak's post-test. **d** For the same mice as in **a**, area under the curve analysis over the time course was used to assess knee hyperalgesia. Two-way ANOVA with Sidak's post-test. (*Piezo2*$^{CKOfl/+}$ vehicle vs. *Piezo2*$^{CKOfl/+}$ NGF: *p* = 0.1243). **e** RNA was extracted from the L3-L5 DRGs of a separate cohort of mice after 3 injections of NGF or vehicle and qPCR was performed (*n* = 3 WT mice/group; *n* = 5 *Piezo2*$^{CKOfl/+}$ mice/group). Unpaired two-tailed *t*-test. Mean ± SEM. DRG dorsal root ganglia, NGF nerve growth factor, VEH vehicle. For **a–e**, source data are provided as a Source Data file.

understand the functional implications in humans. In addition, while we demonstrate an effect on mechanical allodynia of the hind paw associated with aging, we were limited in the joint-specific assessments that could be made in these animals since osteoarthritis damage develops in both hind legs with age. Finally, while the model used here to study the effects of NGF in the knee joint uses injections of NGF that are higher than those found in the human osteoarthritis joint (2–6 ng/mL)[64], this is necessary in order to overcome the initial clearance of the protein[65] following each injection.

To translate this work, development of specific therapeutic strategies to inhibit Piezo2 will be critical. A dietary fatty acid was recently shown to inhibit Piezo2 when activated by bradykinin in vitro;[29] however, the specificity of such approaches is still unclear. In osteoarthritis, intra-articular or topical therapies are attractive options for local delivery, which could avoid unwanted systemic effects of Piezo2 blockade associated with blocking proprioception[2,66]. It has been reported that intra-articular injection of the non-specific mechanosensitive channel inhibitor, GsMTx4, was able to transiently inhibit joint pain in mice[67]. Additionally, intra-articular delivery of Gi-DREADD via AAV was also shown to be effective in reducing inflammatory joint

pain[68]. Importantly the observation that inactivation of a single *Piezo2* allele in mice reduces pain associated with NGF has translational relevance since this suggests that drugs that only partially block PIEZO2 may be effective and would thereby reduce problems associated with complete inhibition of PIEZO2 function.

## Methods

### Study design

The objective of this study was to test the role of Piezo2 expressed by nociceptors in mechanical pain-related behaviors associated with experimental joint pain. To do this we used four different model systems: single intra-articular injection of CFA, destabilization of the medial meniscus surgery, spontaneous osteoarthritis associated with aging, and long-term repeated intra-articular injection of NGF, and we tested the development of pain-related behaviors in control mice and mice with conditional *Piezo2* depletion in nociceptors. In addition, the expression pattern of *Piezo2* and the NGF receptor, *Ntrk1*, in nociceptors was examined in both mouse and human DRGs by RNAscope; additionally, scRNAseq was used to examine mouse DRG expression patterns. For all experiments, the number of biological replicates and

statistical test used are reported in the figure legends. For in vivo experiments, cages of mice were randomly assigned to treatment groups. Behaviors were assessed by an individual blinded to mouse genotype or experimental group. Sample sizes were chosen based on previous experience and publications in our laboratory. All animal experiments were approved by the Institutional Animal Care and Use Committees at Rush University Medical Center and Northwestern University. All human experiments were approved by the Institutional Review Board of Rush University Medical Center.

### Animals

A total of 171 mice were used. Animals were housed with food (Teklad Global 18% Protein Rodent Diet, Inotiv #2018) and water *ad libitum* and kept on 12-h light cycles (light from 7am-7pm). Mice were maintained in HEPA filtered ventilated micro-isolator racks using polycarbonate cages that have 75 in$^2$ of floor space. Mice were housed using Shepherd's® Cob PLUS™ (Shepherd Specialty Papers). Housing rooms were maintained at an average 45% humidity and 72 °F temperature. Na$_V$1.8-Cre mice were obtained as a gift from Dr. John Wood (on C57BL/6 background)[23]. tdTomato loxp (Jax # 007909) or GCaMP6s loxp (Jax # 028866) mice were ordered from Jackson labs and crossed with Na$_V$1.8-Cre mice. Piezo2-Cre (Jax # 027719) and Piezo2 loxp (Jax # 027720) mice were ordered from Jackson labs and crossed with Na$_V$1.8-Cre, Na$_V$1.8-Cre;tdTomato or Na$_V$1.8-Cre;GCaMP6s mice. hM4Di-loxp mice (inhibitory DREADD receptor) (termed Pdi mice) were obtained as a gift from Dr. Susan Dymecki[69]. Piezo2-Cre mice were crossed with Pdi mice to generate heterozygous Piezo2-Pdi mice, used for all chemogenetic experiments. Wild-type C57BL/6J mice were ordered from Jackson labs. All mice used were on a C57BL/6 background. All mice were euthanized under carbon dioxide or isoflurane (1.5–2% in O$_2$).

### Humans

Human DRGs came from participants in the Religious Orders Study (ROS) or Rush Memory and Aging Project (MAP)[70,71]. At enrollment, participants agreed to annual clinical evaluation and organ donation at death, including brain, spinal cord, nerve, and muscle. Both studies were approved by the Institutional Review Board of Rush University Medical Center. All participants signed an informed consent, Anatomic Gift Act, and a repository consent to allow their resources to be shared. The DRGs were removed postmortem and flash frozen as part of the spinal cord removal. DRGs were from three donors (male *n* = 1, female *n* = 2, ages 82-94, BMI, 21.2-27.6). ROSMAP resources can be requested at https://www.radc.rush.edu.

### In vivo DRG calcium imaging

Naive adult male mice, aged 35-39 weeks (*n* = 5 Na$_V$1.8;GCaMP6s$^{fl/+}$; *n* = 4 Na$_V$1.8;GCaMP6s$^{fl/+}$;Piezo2$^{fl/+}$) were used. All mice were deeply anesthetized using isoflurane (1.5–2% in O$_2$), a laminectomy from vertebrae L2-L6 was performed, and the right-side L4 DRG was exposed[18]. This DRG contains cell bodies of sensory neurons that innervate the mouse knee joint and hind paw[72,73]. Silicone elastomer (World Precision Instruments) was used to cover the exposed DRG and surrounding tissue to avoid drying[74]. The mouse was positioned under a Prairie Systems Ultima In Vivo two photon microscope on a custom stage, using Narishige spinal clamps to slightly elevate the mouse in order to avoid motion artifacts associated with breathing. A Coherent Chameleon-Ultra2 Ti:Sapphire laser was tuned to 920 nm and GCaMP6 signal was collected by using a bandpass filter for the green channel (490–560 nm). Image acquisition was controlled using PrairieView software version 5.3 or version 5.5. Images of the L4 DRG were acquired at 0.7 Hz, with a dwell time of 4 μs/pixel (pixel size 1.92 × 1.92 μm$^2$), and a 10× air lens (Olympus UPLFLN U Plan Fluorite, 0.3 NA, 10 mm working distance). The scanned sample region was 981.36 × 981.36 μm$^2$. Anesthesia was maintained using isoflurane

(1.5–2%) during imaging. Mechanical force was applied to the mouse limb using a calibrated forceps device[18]. For each mouse, baseline images were acquired in the absence of force (30 frames), followed by imaging while forces of 30 g or 100 g were applied to the knee joint of the mouse. In addition, dynamic brush was applied to the hind paw using a paint brush. For each stimulus, baseline images were captured for 10 frames prior to the application of the stimulus, the stimulus was applied for 10 frames, and an additional 10 frames were captured after the stimulus was discontinued. Between each stimulus, the mouse was allowed to recover for at least 3 min in order to ensure that all previous neuronal responses had ceased and the fluorescence levels had returned to baseline. In addition, as a positive control, neuronal responses to a 200 g force applied to the ipsilateral hind paw were confirmed[18] prior to proceeding with the rest of the experiment. For each mouse, changes in [Ca$^{2+}$]$_i$ were quantified using a custom ImageJ macro (https://mskpain.center/download_file/8dd82cc8-6c1c-4e37-b466-279afabd6307/530) to calculate the change in fluorescence in each frame t of a time series using the formula: ΔF/Fo, where Fo is the average intensity of the baseline period acquired in the absence of force (30 frames)[18]. Sensory neuron responses to either 30 g or 100 g mechanical force applied to the knee joint or to dynamic brush applied to the hind paw using a paint brush were identified as cells having peak ΔF/Fo during the application period that was greater than 4 times the standard deviation of the baseline period[74]. The total number of neurons imaged for each DRG was estimated by counting the number of neurons within a region of average density and extrapolating to the total imaged area (mean ± SEM: Na$_V$1.8;GCaMP6s = 360 ± 42 neurons; Na$_V$1.8;GCaMP6s;Piezo2$^{fl/+}$ = 294 ± 40 neurons). The percentage of responses to each stimulus was calculated using the formula: # responses / # total neurons imaged × 100. The peak area under the curve (AUC) of each neuron responding to each stimulus was quantified using GraphPad Prism v9 and averaged for each mouse. Cell areas were calculated in Fiji 2.9.0 from each ROI.

### Complete Freund's adjuvant injection study

Under isoflurane anesthesia (1-1.5% in O$_2$), CFA (Millipore, 344289) (5 μg in 5 μL) was injected intra-articulary in the right knee of female *Piezo2*$^{CKOfl/fl}$ (*n* = 7) and female wild-type control (*n* = 5) mice at 20 weeks of age. Prior to injection and at 2, 4, 7, 14, and 21 days post injection, mice were assessed for knee hyperalgesia in the morning, and knee swelling was assessed in the afternoon using a microcaliper. Knee swelling for each mouse was calculated as a difference score for each time point: post-injection - baseline.

### Surgery

DMM surgery was performed in the right knee of 10-week-old male mice (25–30 g) under isoflurane anesthesia[35,75]. After medial para-patellar arthrotomy, the anterior fat pad was dissected to expose the anterior medial meniscotibial ligament, which was severed. The knee was flushed with saline and the incision closed. Mice were not administered analgesia *post* surgery.

### Hind paw mechanical allodynia

Control and *Piezo2*$^{CKO}$ male mice were tested for secondary mechanical allodynia of the ipsilateral hind paw using von Frey fibers and the up-down staircase method[35]. The threshold force required to elicit withdrawal of the paw (median 50% withdrawal) was determined on each hind paw on each testing day. For the DMM experiment corresponding to Fig. S5: No Cre (*n* = 7), *Piezo2*$^{CKOfl/+}$ (*n* = 7) and *Piezo2*$^{CKOfl/fl}$ (*n* = 10) male mice were tested. For the DMM experiment corresponding to Fig. S6: Na$_V$1.8-Cre (*n* = 5) and homozygous *Piezo2*$^{CKOfl/fl}$ (*n* = 6) male mice were tested. Withdrawal thresholds were assessed before surgery and up to 16 weeks after DMM. For the aging experiment corresponding to Fig. 4C, S7: naive no Cre (*n* = 5) and naive *Piezo2*$^{CKOfl/fl}$ male mice (*n* = 8) were tested at 18 months of age.

## Knee hyperalgesia

Knee hyperalgesia was measured using a Pressure Application Measurement (PAM) device (Ugo Basile, Varese, Italy)[33]. Mice were restrained by hand and the hind paw was lightly pinned to make the knee flexion at a similar angle for each mouse. The PAM transducer was pressed against the medial side of the knee and pressure applied against the knee. PAM software guided the user to apply a constantly increasing force (30 g/s) up to a maximum of 450 g. If the mouse tried to withdraw its knee, the force at which this occurred was recorded. Two measurements were taken and recorded per knee by an experimenter blinded to the mouse strain and the withdrawal force data were averaged. If mice did not withdraw their knee, a force of 450 g was assigned. For the DMM experiment corresponding to Fig. S6: Knee hyperalgesia was assessed 4 and 8 weeks after DMM surgery in $Na_V1.8$-Cre$^{+/-}$ ($n = 5$) and $Piezo2^{CKOfl/fl}$ ($n = 6$) male mice. For the DMM experiment corresponding to Fig. 4: wild-type ($n = 5$), and $Piezo2^{CKOfl/fl}$ ($n = 5$) male mice were tested pre-surgery, and 4, 8, 12 and 16 weeks after surgery (these mice were also assessed for weight-bearing asymmetry two days before each knee hyperalgesia test, as described below).

## Weight-bearing asymmetry

Weight bearing asymmetry was assessed utilizing a custom voluntarily accessed static incapacitance (VASIC) method where mice were trained to perform a string-pulling task[38] while freely standing on a static incapacitance meter platform. To accustom mice to the task, a training phase and a testing phase were necessary. The night before each phase, food was removed to increase motivation to the task. Mice were trained to the task in a cage void of stimuli other than 20 strings of varying lengths (53–75 cm) hanging from the wire cage top. Ten of the strings were baited with half of a Honey Nut Cheerio™ to serve as a reward. During the training phase, mice were given one hour to pull all 20 strings, and mice unable to pull at least 15 were retrained. The testing phase occurred the following week in which mice were placed in a custom-built plexiglass chamber atop a Bioseb Static Incapacitance meter (Harvard Apparatus). Weight bearing was recorded once mice had both hind limbs placed on each load cell and were pulling a hanging string (100 cm) to receive a Cheerio reward. Three readings were taken per animal and averaged. Weight bearing was assessed pre-surgery and 4, 8, 12, and 16 weeks post DMM surgery in male wild-type or $Piezo2^{CKOfl/fl}$ mice ($n = 5$ per condition). One week prior to each testing time point, mice were retrained to the task.

## Inhibitory DREADD

Clozapine-N-oxide (CNO) (10 mg/kg in saline, i.a.) (VDM Biochemicals, Inc.) or vehicle (phosphate buffered saline (PBS)) was administered to test the effect of neuronal inhibition on knee hyperalgesia in Piezo2-Pdi male mice 9 weeks after DMM surgery ($n = 10$ vehicle; $n = 9$ CNO). Mice were acclimated to testing 8 weeks after DMM surgery. One week later, when mice were 9 weeks *post* DMM surgery, baseline knee hyperalgesia was tested, and the next day CNO or saline was injected after which knee hyperalgesia was assessed 1, 2 and 4 h *post* injection by a blinded observer.

For analysis of DREADD expression, HA-tag antibody immunofluorescent staining was performed using DRGs collected from male wild-type ($n = 2$, 12 weeks old), $Na_V1.8$-Pdi ($n = 1$, 8 weeks old) and Piezo2-Pdi mice ($n = 6$, 22–31 weeks old). DRGs were fixed in 4% paraformaldehyde (PFA), transferred to 30% sucrose solution for cryoprotection, embedded in OCT and cryo-sectioned onto slides at 12 μm and stored at −80 °C. DRG slides were post-fixed in 4% PFA and dehydration performed following 50%, 75% and two 100% ethanol washes for 5 min each. Sections were permeabilized with 0.5% TritonX-100 and blocked with 5% BSA + 0.1% TritonX-100. Alexa Fluor 488 conjugated anti-HA.11 Epitope Tag antibody (Biolegend catalog #: 901509)[76] (1:100) was loaded in 1% BSA + 0.1% TritonX-100 overnight. Slides were washed and mounted with Vectashield containing DAPI. Imaging was performed as described below.

A control experiment was performed to test the effect of CNO in mice not expressing the Pdi inhibitory DREADD receptor. Twelve-week old male Piezo2-cre$^{+/-}$ or Piezo2-cre$^{+/-}$;Pdi$^{fl/+}$ mice were injected with CFA (5 μg in 5 μL) in the right knee as above ($n = 4$ mice/group). On day 4 after the injection, knee swelling was assessed by micro-caliper and knee hyperalgesia was assessed by PAM device as above. All mice were injected with 10 mg/kg CNO (i.a.) in the right knee as above and one hour later knee hyperalgesia was re-assessed.

## Histopathology of the knee

For the DMM experiment corresponding to Fig. S5: No Cre ($n = 7$), $Piezo2^{CKOfl/+}$ ($n = 7$) and $Piezo2^{CKOfl/fl}$ ($n = 10$) male mice were perfused 16 weeks post DMM, knees were collected and stained with toluidine blue (0.04% w/v)[77]. For the DMM experiment corresponding to Fig. S6: $Na_V1.8$-Cre ($n = 5$) and homozygous $Piezo2^{CKOfl/fl}$ ($n = 6$) mice were taken down 18 weeks after DMM surgery and knees were collected and H&E stained. For the aging experiment corresponding to Fig. 4C, S7: Mechanical allodynia was assessed at 18-months of age and at 22-months of age naive no Cre ($n = 5$) and naive $Piezo2^{CKOfl/fl}$ mice ($n = 8$) were taken down and knees were toluidine blue stained. Knee sections were evaluated for cartilage degeneration and osteophyte width using modified Osteoarthritis Research Society International recommendations[78]. Briefly, for cartilage degeneration: Four joint surfaces, medial and lateral femoral condyles and tibial plateaux, were scored for severity of cartilage degeneration. For each cartilage surface, scores were assigned individually to each of three zones (inner, middle, outer) on a scale of 0–5, with 5 representing the most damage. The maximum score for the sum of femoral and tibial cartilage degeneration on either the medial or lateral side = 30. The maximum possible total cartilage degeneration score for the whole joint (sum of medial and lateral sides) is 60. The largest osteophyte (medial tibia or femur) was measured using an ocular micrometer. Synovial changes were characterized with respect to inflammation type and degree if present (inflammation and/or fibrosis) on a scale of 0–4 (0 = not present, 1 = minimal, 2 = mild, 3 = moderate, or 4 = severe).

## scRNAseq

L3-L5 DRGs were isolated and pooled from 10 C57BL/6 male naive mice at 18 weeks of age and treated with 20 mg/mL collagenase for 15 min. This was followed by addition of digestion mix (1X TrypLE, 25 U/mL Papain, 1 mM DNase1 and 20 mg/mL collagenase/dispase) for 20 min. The cells were pipetted using a regular bore glass pipette for about ten times until no visible clumps remained. The cell suspension was then passed through a 40 μm filter and the filtrate was centrifuged at 100 g for 5 min at 4 °C. The cell pellet was resuspended in 0.5 mL aCSF (87 mM NaCl, 2.5 mM KCl, 1.25 mM $NaH_2PO_4$, 26 mM $NaHCO_3$, 75 mM Sucrose, 20 mM Glucose, 1 mM $CaCl_2$, 7 mM $MgSO_4$) and 0.5 mL DRG media and gently layered on top of an Optiprep gradient (90 μL Optiprep, 455 μL aCSF and 455 μL DRG media). The gradient with the cell suspension was centrifuged at 100 g for 10 min at 4 °C and the process was repeated twice to remove any contaminating myelin debris. The pellet was then resuspended in 100 μL of media along with 10 μL DNase.

Cell number and viability were analyzed using a Nexcelom Cellometer Auto2000 with AOPI fluorescent staining method (92% viability). 9400 naive cells were loaded into the Chromium Controller (10X Genomics, PN-120223) on a Chromium Next GEM Chip G (10X Genomics, PN-1000120), and processed to generate single cell gel beads in the emulsion (GEM) according to the manufacturer's protocol. The cDNA and library were generated using the Chromium Next GEM Single Cell 3' Reagent Kits v3.1 (10X Genomics, PN-1000286) and Dual Index Kit TT Set A (10X Genomics, PN-1000215) according to the manufacturer's manual. Quality control for the constructed library was

performed by Agilent Bioanalyzer High Sensitivity DNA kit (Agilent Technologies, 5067-4626) and Qubit DNA HS assay kit for qualitative and quantitative analysis, respectively. The library was sequenced on an Illumina HiSeq 4000 sequencer with 2 × 50 paired-end kits using the following read length: 28 bp Read1 for cell barcode and UMI, and 90 bp Read2 for transcript. Sequencing reads were assembled and aligned against the mm10-2020-A mouse reference using Cell Ranger v6.0.0 (10x Genomics). Raw fastq files and the expression count matrix have been deposited on NCBI GEO (accession number GSE198485). The expression count matrix was analyzed using the Seurat v4.0.1 R package[79]. In brief, cells were filtered (nFeature_RNA > 200 and percent.mt <15), resulting in 8,755 cells for analysis. Data were log normalized, scaled, and the 2000 most variable features were identified. Clustering was performed using UMAP (dims 17, resolution = 1). Markers for each cluster were identified using the Wilcoxon rank-sum test integrated in Seurat. Clusters were manually annotated using previous DRG datasets as a guide[21,80]. Cells expressing combinations of *Scn10a*, *Piezo2*, and *Ntrk1* were identified as cells having expression levels of these genes > 0.1.

## DRG RNAscope

Mouse L3-L5 DRG were collected (*n* = 4 no Cre controls; *n* = 3 *Piezo2*^CKOfl/+; *n* = 5 *Piezo2*^CKOfl/fl; *n* = 3 wild-type (*n* = 2 male, *n* = 1 female) mice), fixed in 4% paraformaldehyde (PFA), transferred to 30% sucrose solution for cryoprotection, embedded in OCT and cryo-sectioned onto slides at 12 μm. Human DRG (*n* = 1 male, *n* = 2 female) were removed postmortem and flash frozen. DRGs were embedded in OCT and cryo-sectioned onto slides at 16 μm. During sectioning, both mouse and human slides were kept within the cryostat at −20 °C before storage at −80 °C.

RNA in situ hybridization (ISH) was performed using ACD Bio-Techne RNAscope Multiplex Fluorescent v2 Assay. For mouse DRG, manufacturer's instructions were followed. For human DRGs, modifications were made to the protocol to preserve tissue integrity. Briefly summarized, slides were removed from −80 °C and immediately submerged in 4% PFA on ice for 40 min. Dehydration was performed following 50%, 75% and two 100% ethanol washes for 5 min each. Hydrogen peroxide (3%) was applied for 10 min. Target retrieval was performed, reducing time in target retrieval buffer to 3 min followed by protease III incubation for 30 min. The remainder of the protocol was performed following manufacturer's instructions. Probes were used at 1:50 dilution and Opal dyes from Akoya Biosciences were used at 1:100 dilution. A combination of opal dyes 520 (OP-001001), 570 (OP-001003) and 650 (OP-001005) were used. For mouse DRG, *Scn10a* (426011-C2 (probe for the gene encoding Na_V1.8)), *Piezo2-E43-E45* (439971-C3) and *Ntrk1* (435791-C1) probes were used and ACD Bio-Techne DAPI. For human DRG, *SCN10A* (406291-C3 (probe for the gene encoding Na_V1.8)), *PIEZO2* (449951-C2) and *NTRK1* (402631-C1) probes were used. For DAPI staining, Vectashield containing DAPI was used. ACD Bio-Techne positive and negative control probes were conducted for each species prior to start of work. Negative controls were included on every slide.

All imaging was performed using a Fluoview FV10i confocal microscope at ×10 and ×60 magnification (Olympus Fluoview FV10-ASW Ver.04.02). Multiple planes of focus were captured, but Z-stacks were not produced and instead the optimally focused image was chosen for processing and analysis. Laser intensity was used at ≤9.9% throughout. Images were processed and quantified using Fiji software v2.9.0. Only brightness and contrast tools were used to adjust images, and these adjustments have been applied to the entire image, and have been applied across all images and controls. Images were captured at 1024 × 1024 resolution and downstream processing was not performed to enhance the resolution of the images. For mouse DRGs, 2–4 sections per mouse were quantified and averaged. For total subset quantification and diameter analysis, all neurons were outlined to create Regions of Interest (ROI). ROIs were then measured for diameter size, and individually assessed for expression of probes of interest. For human DRGs, 3 images per section over 2–3 sections for each DRG were quantified and averaged. The total number of neurons assessed is indicated in figures. For stitching of human DRG images in Supplemental Figs. 11 and 12, Fiji 'Stitching' plugin was used[81].

## NGF intra-articular injections

Study 1: Recombinant murine NGF (500 ng R&D Systems cat. No.1156) or vehicle (0.1% BSA in PBS) in 5 μL was injected intra-articularly twice a week for 8 weeks into the right knee of naive wild-type or heterozygous *Piezo2*^CKOfl/+ (Na_V1.8-Cre^+/−;GCaMP6s loxp^fl/+;Piezo2 loxp^fl/+ mice) male mice 15–16 weeks old (*n* = 5/group-total of 20 mice). Knee swelling was assessed once before injections started (baseline) and before each injection using a microcaliper. Knee hyperalgesia was assessed at baseline, and at 2, 4 and 8 weeks in a blinded way as described above. Study 2: Recombinant murine NGF (500 ng R&D Systems cat. No.1156) or vehicle (0.1% BSA in PBS) in 5 μL was injected intra-articularly twice a week for a total of 3 injections into the right knee of naive wild-type or *Piezo2*^CKOfl/+ (Na_V1.8-Cre^+/−;GCaMP6s loxp^fl/+;Piezo2 loxp^fl/+ mice) male mice 15–16 weeks old (*n* = 3–5 mice/group). Ipsilateral L3-L5 DRG were collected from mice 24 h after the last injection of NGF or vehicle. Gene expression of *Calca* was analyzed in the DRG cells using qRT-PCR. Briefly, total RNA was extracted using RNeasy mini kit (Qiagen, Hilden, Germany). Reverse transcription was performed using RT$^2$ first strand kit (Qiagen). Quantification of mRNA was conducted using the SYBR Green qPCR master mix (Qiagen) and the RT$^2$ *Calca* primer assay (Qiagen: 330001; GeneGlobe ID: PPM25093C-200) on a Bio-Rad CFX96 machine. The comparative ΔCT method was utilized for relative quantitation of gene levels of expression. *Gapdh* (Qiagen: 33001; GeneGlobe ID: PPM02946E-200), was used as an internal control for normalization of target gene expression.

## Statistics

For RNAscope, counts were compared between genotypes by unpaired two-tailed *t*-test. Distributions were compared using the two-tailed Kolmogorov–Smirnov test. For in vivo calcium imaging, the number of sensory neuron responses to a mechanical stimulus were compared between strains of mice by unpaired two-tailed *t*-test. For mechanical allodynia data, paw withdrawal thresholds were log-transformed prior to further analyses[82]. For mechanical allodynia and knee hyperalgesia time courses, a repeated measures two-way ANOVA with Sidak post-test was used to compare Na_V1.8-Cre and *Piezo2*^CKO responses at each time point. For mechanical allodynia in aged mice, genotypes were compared by unpaired two-tailed *t*-test. For knee hyperalgesia analgesic time courses, a repeated measures two-way ANOVA with Sidak post-test was used to compare mice treated with vehicle to mice treated with CNO at each time point. For knee histopathology, continuous data were analyzed by unpaired two-tailed *t*-test or one-way ANOVA with Sidak post test and non-continuous data were analyzed by two-tailed Mann–Whitney test or Kruska–Wallis followed by Dunn's post test. All analyses were carried out using Graph-Pad Prism version 9 for Mac (GraphPad Software, San Diego, CA). Results are presented as mean ± SEM. Sample size, *p* value, and statistical tests are indicated in the figure legends.

## Reporting summary

Further information on research design is available in the Nature Portfolio Reporting Summary linked to this article.

## Data availability

Single cell RNAseq data have been deposited in NCBI GEO under accession number GSE198485. A reporting summary for this article is available as a Supplementary Information file. The source data

underlying Figures and Supplementary Figures are provided in the Source Data file. Source data are provided with this paper.

## Code availability

Custom ImageJ macro to calculate $\Delta F/Fo$ can be found in the Supplementary Information file or downloaded here (https://mskpain.center/download_file/8dd82cc8-6c1c-4e37-b466-279afabd6307/530).

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

## Acknowledgements

This work was supported by the Northwestern University NUSeq Core Facility. The 10x Genomics Chromium System employed for the scRNA-seq is made available with an NIH S10 Grant to NUSeq (1S10OD025120). Thank you to Dr. Dale George for assistance with optimizing the DRG cell preparation for scRNAseq. We thank the study participants and staff of the Rush Alzheimer's Disease Center. ROSMAP resources can be requested at https://www.radc.rush.edu. We would also like to acknowledge these funding sources: National Institutes of Health grant R01AR077019 (R.E.M.), National Institutes of Health grant R01AR064251 (A.M., R.J.M.), National Institutes of Health grant R01AR060364 (A.M.), National Institutes of Health grant P30AR079206 (A.M.), Rheumatology Research Foundation Innovative Research Award (A.M.), National Institutes of Health grants P30AG10161, P30AG72975, R01AG15819, and R01AG17917 (D.A.B.).

## Author contributions

Conceptualization: A.M.O., M.J.W., R.J.M., A.M., R.E.M. Methodology: A.M.O., M.J.W., N.S.A., S.I., L.W., D.R., R.J.M., A.M., R.E.M. Software: R.E.M. Investigation: A.M.O., M.J.W., N.S.A., S.I., J.L., L.W., D.R., R.E.M., Resources: D.A.B. Visualization: A.M.O., M.J.W., R.E.M. Funding acquisition: R.J.M., A.M., R.E.M. Project administration: R.J.M., A.M., R.E.M. Supervision: R.J.M., A.M., R.E.M. Writing—original draft: A.M.O., M.J.W., R.J.M., A.M., R.E.M. Writing—review & editing: A.M.O., M.J.W., N.S.A., S.I., J.L., L.W., D.R., D.A.B., R.J.M., A.M., R.E.M.

## Competing interests

The authors declare the following competing interests: A.M. M. is a consultant of Asahi Kasei Pharma and a consultant of 23andMe. The remaining authors declare no competing interests.
