## [Peer Review File · Nature Communications]

Piezo2 expressing nociceptors mediate mechanical sensitization in experimental osteoarthritisReviewers' Comments:

Reviewer #1:

Remarks to the Author:

The paper "Piezo2 expressing ..." by Obeidat and colleagues reports new and interesting findings related to osteoarthritis-related joint pain.

It has strong translational relevance given the large unmet medical need of the disease targeted, painful osteoarthritis.

It is a strong point of the study to make the link to NGF because NGF neutralization with therapeutic antibodies has resolved osteoarthritis pain in patients. The authors' mouse work - which not only features conditional knockout of Piezo2 in Nav1.8-expressing nociceptor DRG neurons, but also their chemogenetic hyperpolarization via intra-articular injection of the activator compound (which is neat !) - is then reassuringly complemented by high-resolution expression studies in human DRGs.

The experiments are overall conducted in straightforward manner, also well-described.

However, what is missing, in order to bring the study full circle in terms of translational relevance is an analysis of the gait disturbance evoked by knee joint injury, and how nociceptor knockdown of Piezo2 impacts (repairs ?) the gait disturbance.

Also, the authors need to provide comment on the amounts of NGF applied intra-articularly and how this compares to NGF concentration measured in human OA joint fluid.

Reviewer #2:

Remarks to the Author:

This manuscript by Obeidat & Wood et al. investigates the role of Piezo2-positive nociceptors in mediating joint pain. There are some great strengths of this manuscript, including varied approaches, that will move the field forward. I believe that this is the first functional evidence that implicates Piezo2 as the mechanotransduction channel in a silent nociceptor population (a continuation of the Prato & Taberner 2017 paper), rather than in the context of mechanical allodynia where it is acting in touch receptors. Even still, there are some confusing omissions and areas that need clarification for this paper to have maximal impact.

Major concerns:

1) The overall messaging is confusing. In Figure 2, the authors show calcium responses depend on Piezo2 for what they refer to "non-noxious" stimuli. This is valuable data but seems to run counter to their primary message in the rest of the manuscript that Piezo2 is critical for nociception during inflammatory or osteoarthritis conditions. Could it not be both? In fact, the title of their Fig 2 legend is self-contradictory: "nociceptor expression of Piezo2 in mediating responses to non-noxious mechanical stimuli". If something is responding to non-noxious stimuli it is not a nociceptor! Also, it would be helpful for a broader audience to describe why these stimuli are not considered noxious in the first place?

2) Given this imaging setup, the authors have a chance to test much more than is shown. This would help clarify their message about what Piezo2 is doing in joint mechanosensation and could address the noxious vs non-noxious distinction. For example, this lab has previously done a knee twisting model, which is certainly noxious. This imaging would be very interesting to see in Piezo2 KO animals.

3) I am confused as to why they use heterozygous mice instead of full knockout mice in Fig 2. It is true that hets seem sufficient to see a change, which is surprising, but it muddies the waters with

regard to what one can really learn because any changes will be partial. Full knockouts were used in later figures. The authors should clarify this point and why this was done.

4) Fig 3: why is A missing times, specifically time 0? It seems like A and B should be done together? Are these the same cohorts? Moreover, it is not clear that 3B and 3C have anything to do with joint pain specifically because they are both assessing paw withdrawal. How do we know if older mice just have different cutaneous sensitivity? To make claims about knee pain, the authors must do weight bearing or some other behavioral assessments.

5) The authors claim importance for inflammation, but only inject NGF. They could do an MIA model to show another inflammation example and bolster this claim. This is particularly relevant in light of the Prato & Taberner paper that shows NGF specifically causes mechanosensitivity in silent nociceptors. Is this a response to any inflammatory model, or specifically downstream of NGF?

Minor concerns

6) Authors switch to "Nav1.8." after using the Scn10a gene name. They should standardize nomenclature throughout.

7) Fig 4: Why did authors apply treatment at 9 weeks after DMM? Different endpoints have been used in different figures without much explanation. E.g. Fig 3, endpoint week 16; sup fig 6, endpoint week 18; sup fig 5, didn't state endpoint. It would be better if author have consistent endpoints throughout or explain why different endpoints are used in different figures within the same study.

8) Did the authors try CNO injection in WT mice to see if there was a drug-specific effect?

REVIEWER COMMENTS

Reviewer #1 (Remarks to the Author):

The paper "Piezo2 expressing ..." by Obeidat and colleagues reports new and interesting findings related to osteoarthritis-related joint pain.

It has strong translational relevance given the large unmet medical need of the disease targeted, painful osteoarthritis.

It is a strong point of the study to make the link to NGF because NGF neutralization with therapeutic antibodies has resolved osteoarthritis pain in patients. The authors' mouse work - which not only features conditional knockout of Piezo2 in Nav1.8-expressing nociceptor DRG neurons, but also their chemogenetic hyperpolarization via intra-articular injection of the activator compound (which is neat !) - is then reassuringly complemented by high-resolution expression studies in human DRGs.

The experiments are overall conducted in straightforward manner, also well-described.

However, what is missing, in order to bring the study full circle in terms of translational relevance is an analysis of the gait disturbance evoked by knee joint injury, and how nociceptor knockdown of Piezo2 impacts (repairs ?) the gait disturbance.

Response: Thank you for the positive feedback on our study. We agree that a non-evoked measurement of pain-related behavior would improve our study. The reviewer suggests analysis of gait – and we agree that changes in gait provide valuable information and are translationally relevant. However, developing a reliable gait system within a laboratory is not trivial (PMID: 26995111), and we are not setup at this time to report gait data in mice. As an alternative, which is also in response to a comment made by reviewer 2, we have added a new experiment where we evaluated weight-bearing asymmetry and knee hyperalgesia concurrently in WT and Piezo2^{CKO/fl/fl} mice for 16 weeks after DMM surgery. We found that WT mice developed weight-bearing asymmetry by 12 weeks after surgery, consistent with prior work, and Piezo2^{CKO/fl/fl} mice were protected from developing this behavior. Concurrently, Piezo2^{CKO/fl/fl} mice developed less knee hyperalgesia compared to WT mice, consistent with our prior experiment that was included in the original manuscript (original Fig 3, now Supp Fig 6). In total we now show 3 independent DMM experiments with a combination of hind paw mechanical allodynia, knee hyperalgesia, and hind leg weight-bearing asymmetry data to support the overall conclusion that knocking-out Piezo2 from nociceptors impacts the development of mechanical sensitization and pain behavior associated with weight-bearing.

Fig. 4. Piezo2 plays a role in mechanical sensitization in two mouse models of osteoarthritis. a Weight-bearing asymmetry was assessed in wild-type (n=5) and *Piezo2*^{CKOfl/fl} (n=5) male mice after DMM surgery. Dashed line indicates 0 g which would represent mice putting equal weight on both hind legs; negative numbers indicate mice are putting more weight on the uninjured left rear leg. Two-way repeated measures ANOVA with Sidak post-test. **b** Knee hyperalgesia was assessed in mice from part **a**. Dashed line indicates maximum of the assay: 450 g. Contralateral unoperated left legs: 4 weeks (mean±SEM): wild-type (425±2) and *Piezo2*^{CKOfl/fl} (437±7). 8 weeks: wild-type (439±2) and *Piezo2*^{CKOfl/fl} (445±5). 12 weeks: wild-type (437±2) and *Piezo2*^{CKOfl/fl} (432±7). 16 weeks: wild-type (441±2) and *Piezo2*^{CKOfl/fl} (446±4). Two-way repeated measures ANOVA with Sidak post-test. An independent experiment is shown in Supp. Fig. 6. **c** Hind paw mechanical allodynia was assessed in naïve control mice (littermate no cre, n=5) or *Piezo2*^{CKOfl/fl} (n=8) at 1.5 years of age. Histology shown in Supp. Fig. 7. Unpaired two-tailed t-test on log-transformed data.

Action: We have modified the original main figure 3 (now main figure 4) to include this new DMM experiment. The DMM experiment that had been shown in the original Fig 3A,B has been moved to Supp Fig 6.

Also, the authors need to provide comment on the amounts of NGF applied intra-articularly and how this compares to NGF concentration measured in human OA joint fluid.

Response: A paper from Dr. Rita Levi-Montalcini's group demonstrated that NGF concentrations of 2-6 ng/mL NGF can be found in human osteoarthritis synovial fluid (PMID: 1536673). Here, we are trying to model the effect of NGF by using bolus exogenous injections twice a week instead of relying on cells in the joint to continually produce NGF over time. Therefore, the amount of NGF injected each time is higher than the amount detected in osteoarthritis synovial fluid since we have to overcome the rapid clearance by the lymphatic system in the joint (PMID: 24189839).

Action: We have added a line in the limitations paragraph of the discussion to comment on how this model compares to the situation in an osteoarthritis joint. Line 405-408 (no track changes version): "Finally, while the model used here to study the effects of NGF in the knee joint uses injections of NGF that are higher than those found in the osteoarthritis joint (PMID: 1536673), this is necessary in order to overcome the initial clearance of the protein (PMID: 24189839) following each injection."

Reviewer #2 (Remarks to the Author):

This manuscript by Obeidat & Wood et al. investigates the role of Piezo2-positive nociceptors in mediating joint pain. There are some great strengths of this manuscript, including varied approaches, that will move the field forward. I believe that this is the first functional evidence that implicates Piezo2 as the mechanotransduction channel in a silent nociceptor population (a continuation of the Prato & Taberner 2017 paper), rather than in the context of mechanical allodynia where it is acting in touch receptors. Even still, there are some confusing omissions and areas that need clarification for this paper to have maximal impact.

Major concerns:

1) The overall messaging is confusing. In Figure 2, the authors show calcium responses depend on Piezo2 for what they refer to “non-noxious” stimuli. This is valuable data but seems to run counter to their primary message in the rest of the manuscript that Piezo2 is critical for nociception during inflammatory or osteoarthritis conditions. Could it not be both? In fact, the title of their Fig 2 legend is self-contradictory: “nociceptor expression of Piezo2 in mediating responses to non-noxious mechanical stimuli”. If something is responding to non-noxious stimuli it is not a nociceptor! Also, it would be helpful for a broader audience to describe why these stimuli are not considered noxious in the first place?

Response: Thank you for pointing out the confusing terminology related to Figure 2. As the reviewer states, even though this is *in vivo* calcium imaging as opposed to calcium imaging on isolated cells, it remains difficult to classify particular forces as noxious or non-noxious since these experiments are being performed in anesthetized animals, and therefore we are analyzing what forces are required to elicit intracellular calcium fluxes instead of what forces are required to elicit behavioral responses that may be interpreted as indicative of pain. Originally, we used these terms based on our behavioral evidence in awake animals that shows forces higher than 400 g need to be applied to the knee joint in order to evoke withdrawal responses in healthy animals. However, we agree that this is confusing and therefore are modifying our description of this experiment to simply indicate our results with respect to particular magnitudes of gram force stimuli.

Action: We have modified the description of the results pertaining to Figure 2 so that noxious / non-noxious terminology is no longer used. We have also modified the title to Figure 2:

Fig. 2. *In vivo* calcium imaging demonstrates a role for nociceptor expression of Piezo2 in mediating intracellular calcium responses to mechanical force applied to the knee joint

2) Given this imaging setup, the authors have a chance to test much more than is shown. This would help clarify their message about what Piezo2 is doing in joint mechanosensation and could address the noxious vs non-noxious distinction. For example, this lab has previously done a knee twisting model, which is certainly noxious. This imaging would be very interesting to see in Piezo2 KO animals.

Response: In response to this comment and to comment 3 below, we attempted to perform additional *in vivo* calcium imaging experiments this past fall, but unfortunately, we were limited by the number of mice available with the correct genotype. Therefore, only three control and two homozygous Piezo2^{CKO/fl/fl} animals were imaged, and we tested the response to 100g force applied to the knee as well as a knee twist. With this number of animals it is not possible to address this comment on the level of the organism as we reported our data in the original Fig 2C,D. However, if we examine this data on the individual neuron level, we can at least look at the sizes of the neurons that responded to each type of stimulus as we reported in the relative frequency plot shown in the original Fig 2E. Qualitatively it appears that the sizes of the neurons still responding to the 100g force stimulus in both the fl/+ and fl/fl Piezo2^{CKO} mice are larger than the neurons in the control mice, and a difference in the distributions is confirmed by the Kolmogorov-Smirnov test (100g control vs. fl/+: p=0.0087; 100g control vs. fl/fl: p=0.0292; 100g fl/+ vs. fl/fl: p=0.35). Likewise, the knee twist provoked responses in larger cells in the fl/fl Piezo2^{CKO} mice compared to the controls (p=0.0102). This suggests that the neurons that still respond to these stimuli in the absence or partial depletion of Piezo2 represent a different type of nociceptor and/or are mediated by a different channel than in the control animals. We agree that this is a very interesting question and may ultimately hint at what other channels could be responsible for mediating responses to high intensity mechanical pain in healthy animals; however, at this point we

estimate that to answer this question in a more detailed manner would take at least 12 months. Therefore, we are showing this frequency distribution plot for reviewer purposes only at this time since we think that this data is not powered sufficiently to add to the manuscript.

Data collected from an additional three control and two $Piezo2^{CKOfI/fl}$ mice was added to the relative frequency plot shown in Fig 2E. For the additional animals imaged, the sizes of the neurons that responded to 100 g of mechanical force applied to the knee or to a noxious knee twist were calculated in Fiji. Kolmogorov-Smirnov test (100g: control vs fl/+ : $p=0.0087$; control vs fl/fl : $p = 0.0292$; fl/+ vs fl/fl : $p=0.35$) (knee twist: control vs fl/fl : $p=0.0102$).

In addition, in the meantime, another relevant paper has been published that used $Trpv1-Cre;Piezo2^{CKOfI/fl}$ mice to study the role of $Piezo2$ in visceral mechanical hypersensitivity (PMID: 36563677). Using this alternative method to target a subset of nociceptors, this study showed that as increasing forces are applied to circumferentially stretch the colon, an increasing number of action potentials are fired in control animals and this is reduced in the $Trpv1-Piezo2^{CKOfI/fl}$ mice, including at the highest forces tested. Together with our results, this study supports the idea that $Piezo2$ expression by nociceptors in different peripheral tissues can contribute to the ability of nociceptors to fire action potentials in response to varying mechanical forces.

Action: We have referenced this new $Trpv1-Cre;Piezo2^{CKOfI/fl}$ paper in the discussion (line 360-369, version without track changes):

“Previous work has shown that *Piezo2* depletion alters C and $A\delta$ nociceptor responses to mechanical stimuli *ex vivo*⁴ and in select subclasses of nociceptors *in vivo*^{20, 21}. In addition, depletion of *Piezo2* from TRPV1-lineage neurons reduced the number of action potentials fired when the colon was circumferentially stretched⁴². Here, by using *in vivo* calcium imaging of the DRG, we also observed that nociceptor responses to 30 or 100 g of mechanical force applied to the knee joint were decreased in anesthetized $Piezo2^{CKOfI/+}$ mice. However, we found that

nociceptor deletion of Piezo2 only impacts behavioral responses of mice after joint damage or inflammation has been initiated. This is consistent with other work suggesting that in the absence of inflammation, high intensity mechanical stimuli are detected by an as yet unidentified channel/receptor^{21, 43, 44.}”

3) I am confused as to why they use heterozygous mice instead of full knockout mice in Fig 2. It is true that hets seem sufficient to see a change, which is surprising, but it muddies the waters with regard to what one can really learn because any changes will be partial. Full knockouts were used in later figures. The authors should clarify this point and why this was done.

Response: Heterozygous mice were used for the experiments in Fig 2 due to practical reasons since it was difficult to generate sufficient numbers of homozygote CKO mice with GCaMP6 present heterozygously. We have since been able to image two such mice. As can be seen in the response to comment 2 above, the sizes of the neurons responding to various stimuli in these mice fell within the variation seen in the heterozygous animal responses, and therefore these results offer some evidence that the reduction of Piezo2 expression in the heterozygous animals is sufficient to alter the Nav1.8+ neuron responses in a manner that is reflective of a full knockout. It should be noted that, as we indicated in the original text, both heterozygous CKO mice and homozygous CKO mice were used in one of the DMM experiments shown in the original Supp Fig 5. In addition, heterozygous CKO mice were used for the NGF experiment in the original Fig 6 (now Fig 7). The phenotype seen in these heterozygous CKO mice would also be consistent with a previous paper that showed an incomplete knockdown of Piezo2 in rats was still sufficient to alter the response of A δ fibers to intraosseus pressure in healthy rats (PMID: 34335288 - Figures 1C and 2C of this paper are pasted here for convenience).

Fig 1C from PMID: 34335288: Densitometry analysis revealed a significant reduction in the ratio of Piezo2/ β -Actin in the DRG of animals injected with Piezo2 antisense ODNs (n = 6) relative to mismatch control ODNs (n = 6). Data represents mean \pm SEM, *P < 0.05, unpaired t-test.

Fig 2C from PMID: 34335288: There was a significantly reduced total discharge frequency in A δ bone afferent neurons recorded from Piezo2 antisense treated animals compared to mismatch control animals [mixed model, F (3.278), DFn (2), Dfd (43), *Dunnett's P < 0.05; n = 23 naïve/31 mismatch/35 antisense, N = 10 naïve/15 mismatch/21 antisense].

This data suggests that future work aimed at identifying humans with a heterozygous mutation in Piezo2 may enable us to learn more about the role of Piezo2 in mechanical hypersensitivity.

Action: Same as comment 2 above.

4) Fig 3: why is A missing times, specifically time 0? It seems like A and B should be done together? Are these the same cohorts? Moreover, it is not clear that 3B and 3C have anything to do with joint pain specifically because they are both assessing paw withdrawal. How do we know if older mice just have different cutaneous sensitivity? To make claims about knee pain, the authors must do weight bearing or some other behavioral assessments.

Response: The knee hyperalgesia and mechanical allodynia experiments were done on the same animals but by two different testers. Since this was during COVID, we were limited in the time points that could be captured due to restrictions in place at the time. In response to this request and a request by reviewer 1, we have since performed a new DMM surgery experiment where all time points were included and both knee hyperalgesia and weight-bearing asymmetry were measured. For aging animals, weight-bearing asymmetry cannot be assessed since OA is developing in both knee joints. Therefore, we have not performed any additional experiments in aged mice, but we have added an additional note regarding this limitation in the discussion.

Fig. 4. Piezo2 plays a role in mechanical sensitization in two mouse models of osteoarthritis. a Weight-bearing asymmetry was assessed in wild-type (n=5) and *Piezo2*^{CKOfl/fl} (n=5) male mice after DMM surgery. Dashed line indicates 0 g which would represent mice putting equal weight on both hind legs; negative numbers indicate mice are putting more weight on the uninjured left rear leg. Two-way repeated measures ANOVA with Sidak post-test. **b** Knee hyperalgesia was assessed in mice from part a. Dashed line indicates maximum of the assay: 450 g. Contralateral unoperated left legs: 4 weeks (mean±SEM): wild-type (425±2) and *Piezo2*^{CKOfl/fl} (437±7). 8 weeks: wild-type (439±2) and *Piezo2*^{CKOfl/fl} (445±5). 12 weeks: wild-type (437±2) and *Piezo2*^{CKOfl/fl} (432±7). 16 weeks: wild-type (441±2) and *Piezo2*^{CKOfl/fl} (446±4). Two-way repeated measures ANOVA with Sidak post-test. An independent experiment is shown in Supp. Fig. 6. **c** Hind paw mechanical allodynia was assessed in naïve control mice (littermate no cre, n=5) or *Piezo2*^{CKOfl/fl} (n=8) at 1.5 years of age. Histology shown in Supp. Fig. 7. Unpaired two-tailed t-test on log-transformed data.

Action: We have modified the original main figure 3 (now Fig 4) to include this new DMM experiment. The DMM experiment that had been shown in the original Fig 3A,B has been moved to Supp Fig 6. We have also added this limitation regarding the aging experiment to the discussion (line 403-405): “In addition, while we demonstrate an effect on mechanical allodynia of the hind paw associated with aging, we were limited in the joint-specific assessments that could be made in these animals since osteoarthritis damage develops in both hind legs with age.”

5) The authors claim importance for inflammation, but only inject NGF. They could do an MIA model to show another inflammation example and bolster this claim. This is particularly relevant in light of the Prato & Taberner paper that shows NGF specifically causes mechanosensitivity in silent nociceptors. Is this a response to any inflammatory model, or specifically downstream of NGF?

Response: We agree that understanding in more detail how inflammation sensitizes Piezo2 could be important translationally, particularly since inflammation is an important contributor to both joint pain and sensitization (PMID: 26554395). While the MIA model promotes knee swelling, it does not reflect the development of osteoarthritis pathologically since the chemical kills the chondrocytes. Therefore, we chose to use a model that has been used to study inflammatory joint pain, the complete Freund's adjuvant (CFA) model (as in PMID: 30240782). For this experiment, we also took the opportunity to use female mice. We found that in this model, where knee swelling occurs very rapidly following the single CFA injection, Piezo2^{CKOfl/fl} mice were not protected from knee swelling, but they were protected from developing accompanying knee hyperalgesia compared to wild-type mice.

Fig. 3. Piezo2 depletion protects from CFA induced mechanical sensitization. (A) Knee swelling was assessed in WT (n=5) or homozygous female Piezo2^{CKOfl/fl} (n=7) mice given a single intra-articular injection of Complete Freund's adjuvant (i.a. 5 μ g, 5 μ L). Two-way repeated measures ANOVA: WT vs. Piezo2^{CKOfl/fl}, interaction p-value = 0.43. (B) Area under the curve analysis over the time course was used to assess knee swelling. Unpaired two-tailed t-test. (C) Knee hyperalgesia was assessed in the same mice as part A. Dashed line indicates maximum of the assay: 450 g. Two-way repeated measures ANOVA with Sidak's post-test. (D) Area under the curve analysis over the time course was used to assess knee hyperalgesia. Unpaired two-tailed t-test.

This is compatible with previous *in vitro* literature from the Patapoutian group showing that bradykinin can sensitize nociceptors in a Piezo2 dependent manner (PMID: 22921401). Additionally, prior work from the Mantyh group showed that anti-NGF antibody treatment could reduce hyperalgesia in a repeated injection CFA model (PMID: 22246649), suggesting that there could still be a mechanistic link involving both the NGF and Piezo2 pathways in this model.

Action: This experiment has been added as a new main Figure 3 with the following results text (line 152-163):

“Depletion of *Piezo2* from nociceptors protects mice from mechanical sensitization associated with acute joint inflammation

Prior *in vitro* work has demonstrated that *Piezo2* activity can be enhanced on short time scales by inflammatory molecules such as bradykinin^{24, 25}. Therefore, we decided to test the role of *Piezo2* in mediating pain associated with acute inflammation in the knee joint following a single injection of complete Freund’s adjuvant (CFA). As others have demonstrated²⁶, injection of CFA caused rapid knee swelling in female wild-type mice (Fig. 3A,B). Accompanying this swelling, wild-type mice developed knee hyperalgesia (Fig. 3C,D), which resolved as the swelling resolved through 21 days after the injection. Female homozygous *Piezo2*^{CKOfl/fl} mice injected with CFA also developed rapid knee swelling (Fig. 3A,B), but they developed less knee hyperalgesia compared to wild-type mice (Fig. 3C,D), suggesting that *Piezo2* may play a role *in vivo* in mediating mechanical sensitization in response to acute joint inflammation.”

We have also added to the discussion (line 370-374):

“Other work has shown that *Piezo2* activity can be enhanced on short time scales by inflammatory molecules such as bradykinin^{24, 25} and NGF^{16, 20}, but exactly which signaling pathways are involved remains unclear^{52, 53, 54}. Here we have shown that this is also relevant *in vivo* by demonstrating that *Piezo2*^{CKOfl/fl} mice have reduced knee hyperalgesia induced by injecting CFA into the knee joint.”

Minor concerns

6) Authors switch to “Nav1.8.” after using the *Scn10a* gene name. They should standardize nomenclature throughout.

Response and Action: We have modified the manuscript to use Nav_v1.8 throughout.

7) Fig 4: Why did authors apply treatment at 9 weeks after DMM? Different endpoints have been used in different figures without much explanation. E.g. Fig 3, endpoint week 16; sup fig 6, endpoint week 18; sup fig 5, didn’t state endpoint. It would be better if author have consistent endpoints throughout or explain why different endpoints are used in different figures within the same study.

Response:

Point 1: In the original Fig 4 (now Fig 5): *Piezo2*-Cre;*Pdi* mice were first acclimated for knee hyperalgesia testing 8 weeks after DMM surgery. The time course test with CNO was performed one week later (9 weeks after surgery). In our experience using this model (>12 years) and assessing knee hyperalgesia (>5 years), mice have similar knee hyperalgesia thresholds 8 and 9 weeks after DMM surgery, and thus we chose to perform the experiment this way in order to best perform the acclimatization step. This has been clarified in the methods.

Point 2: Because most experiments were performed during COVID, the timing of different outcome measures as well as the take down of animals had to be adjusted to accommodate laboratory schedules. We have improved the labeling and re-arranged figures so that it is more clear which plots belong to the same set of animals and what the take down time point was. In general, because these are long models,

we view it as a strength that similar trends can be seen with slightly different time point assessments in multiple independent experiments.

Experiment 1 DMM: Supp Fig 5: hind paw allodynia assessed pre surgery and 4-16 weeks after surgery; mice taken down 16 weeks after surgery and joints processed for histology

Experiment 2 DMM: Supp Fig 6: (this now also contains the plots shown in the original Fig 3A,B): hind paw allodynia assessed pre surgery and 4, 8, 12 and 16 weeks after surgery, knee hyperalgesia was assessed 4 and 8 weeks after surgery; mice were taken down 18 weeks after surgery and joints processed for histology

Experiment 3 DMM (newly added): Fig 4A,B: This experiment has been newly performed as explained above – knee hyperalgesia and weight-bearing asymmetry were assessed pre-surgery, and 4, 8, 12 and 16 weeks after surgery.

Experiment 4 Aging: Fig 4C (original Fig 3C), Supp Fig 7: hind paw allodynia was assessed when mice were 18 months old and these animals were taken down to assess development of joint damage when they were 22 months old.

Action:

Point 1:

Methods section:

“**Inhibitory DREADD:** Clozapine-N-oxide (CNO) (10 mg/kg in saline, i.a.) (VDM Biochemicals, Inc.) or vehicle (phosphate buffered saline (PBS)) was administered to test the effect of neuronal inhibition on knee hyperalgesia in Piezo2-Pdi mice 9 weeks after DMM surgery (n=10 vehicle; n=9 CNO). Mice were acclimated to testing 8 weeks after DMM surgery. One week later, when mice were 9 weeks *post* DMM surgery, baseline knee hyperalgesia was tested, and the next day CNO or saline was injected after which knee hyperalgesia was assessed 1, 2 and 4 hours *post* injection by a blinded observer.”

Point 2:

We have modified the methods, figures, and the figure legends listed above to improve clarity.

8) Did the authors try CNO injection in WT mice to see if there was a drug-specific effect?

Response: We have previously shown that i.p. injection of CNO had no effect on behaviors in wild-type mice (PMID: 28380690, supp figure 2). However, to address the i.a. route of administration used in this study with a more concentrated formulation of CNO in a small volume, we have added a new experiment using the CFA inflammatory knee pain model. Piezo2-cre or Piezo2-cre;Pdi male mice were injected with CFA in the right knee. Four days later, all mice had developed knee hyperalgesia (day 4 CFA in the plot). All mice were injected with 10mg/kg CNO, i.a. and one hour later only Piezo2-cre;Pdi mice had reduced knee hyperalgesia (1 hour post CNO).

Action: We have added this plot into Supp Fig 8.

Reviewers' Comments:

Reviewer #1:

Remarks to the Author:

The authors have conducted an additional functional and translationally-relevant measurement of a joint pain equivalent, weight-bearing asymmetry, which - after some deliberation - addresses my question. I was requesting a gait analysis as a straightforward requirement when conducting a study on OA pain. Aiming for a gait analysis of their mouse model should remain a high priority for the authors for future studies, however they intend to realize this.

Importantly, the authors now clarify the issue of quantity of NGF injected into the joint and resulting NGF concentration intra-articular vs joint concentrations of NGF in osteoarthritis pain. This clarification is important as it means that their current investigation finds its strengths in basic-mechanistic elucidation rather than in translational-medical relevance. For the latter, intra-articular NGF concentrations in model systems of osteoarthritis pain will have to mimic NGF exposures in the human disease, which their current study does not validly recapitulate. The authors need to update and extend their "Limitations of this study" accordingly and explicitly state differences in NGF concentration intra-articular in their model vs what is known in human disease. Also, this issue directly implies that the conclusion drawn toward the end of their abstract, and all respective references to it in the main ms., need to be toned down to the equivalent of "These results SUGGEST that Piezo2 plays a POSSIBLE key role in nociceptor sensitization processes in the osteoarthritic joint, and targeting Piezo2 may represent a novel therapy for osteoarthritis pain control." whichever way the authors finally end up phrasing it.

Reviewer #2:

Remarks to the Author:

The revised manuscript by Obeidat & Wood et al. is fascinating and really adds to the field's knowledge about the role of Piezo2 in mechanical nociception in the context of inflammation. I am impressed by the new experimental models and weight bearing measurement data - they really add interesting and valuable information. I feel that this revised version more strongly supports the author's claims, and has cleared up some of the previously confusing terminology.

My remaining small suggestion is to adjust the data visualization for the calcium imaging in Figure 2. The writing in the heatmap area is quite small. Moreover, the labeling of the start and end of stimuli in different frames is missing from the WT condition.

I am still surprised by the heterozygous results, but the data are clear!

I thank the authors for this nice manuscript and valuable contribution to the field.

Response to Reviewers

REVIEWERS' COMMENTS

Reviewer #1 (Remarks to the Author):

The authors have conducted an additional functional and translationally-relevant measurement of a joint pain equivalent, weight-bearing asymmetry, which - after some deliberation - addresses my question. I was requesting a gait analysis as a straightforward requirement when conducting a study on OA pain. Aiming for a gait analysis of their mouse model should remain a high priority for the authors for future studies, however they intend to realize this.

Importantly, the authors now clarify the issue of quantity of NGF injected into the joint and resulting NGF concentration intra-articular vs joint concentrations of NGF in osteoarthritis pain. This clarification is important as it means that their current investigation finds its strengths in basic-mechanistic elucidation rather than in translational-medical relevance. For the latter, intra-articular NGF concentrations in model systems of osteoarthritis pain will have to mimic NGF exposures in the human disease, which their current study does not validly recapitulate. The authors need to update and extend their "Limitations of this study" accordingly and explicitly state differences in NGF concentration intra-articular in their model vs what is known in human disease. Also, this issue directly implies that the conclusion drawn toward the end of their abstract, and all respective references to it in the main ms., need to be toned down to the equivalent of "These results SUGGEST that Piezo2 plays a POSSIBLE key role in nociceptor sensitization processes in the osteoarthritic joint, and targeting Piezo2 may represent a novel therapy for osteoarthritis pain control." whichever way the authors finally end up phrasing it.

Author response:

We should clarify that the idea that NGF signaling is involved in the pathogenesis of osteoarthritis pain is already very well established based on both human (for review, PMID: 33219344) and pre-clinical studies (PMID: 22246649, 20350782, 27208420, 30862648). Indeed, these studies served as the basis for large scale clinical trials testing the efficacy of anti-NGF on osteoarthritis pain (for review, PMID: 33219344). This program was stopped due to adverse effects related to rapidly progressive osteoarthritis in ~7% of participants, but nonetheless, analgesic efficacy was clearly demonstrated with anti-NGF.

Here we present four translationally-relevant experimental knee pain models (CFA inflammatory knee pain, DMM surgery induced osteoarthritis, aging related spontaneous osteoarthritis, and NGF injection induced chronic knee pain). Indeed, the CFA model and translationally-relevant surgical pre-clinical models of osteoarthritis have already previously been shown to generate pain that is dependent on NGF (PMID: 22246649, 20350782, 27208420, 30862648). Therefore, we chose these models in order to build on that work to demonstrate that joint pain is also linked to Piezo2 signaling by using nociceptor-specific Piezo2

conditional knock-out mice. These mice showed reduced mechanical sensitization behavior in all 4 models of knee pain. We have updated the abstract, introduction and discussion to provide additional information on the known link between OA pain and NGF signaling. We also provide the synovial fluid levels of NGF in human OA joints in the limitations paragraph of the discussion.

Author action:

We have added additional information on the extensive literature demonstrating that different models of joint pain rely on NGF signaling to more clearly illustrate that the role of NGF in osteoarthritis pain is a very well established idea in the field, and the novelty of this study is relating this process to Piezo2.

Abstract: Non-opioid targets are needed for addressing osteoarthritis pain, which is mechanical in nature and associated with daily activities such as walking and climbing stairs. Piezo2 has been implicated in the development of mechanical pain, but the mechanisms by which this occurs remain poorly understood, including the role of nociceptors. Here we show that nociceptor-specific *Piezo2* conditional knock-out mice were protected from mechanical sensitization associated with inflammatory joint pain in female mice, joint pain associated with osteoarthritis in male mice, as well as both knee swelling and joint pain associated with repeated intra-articular injection of nerve growth factor in male mice. Single cell RNA sequencing of mouse lumbar dorsal root ganglia and *in situ* hybridization of mouse and human lumbar dorsal root ganglia revealed that a subset of nociceptors co-express *Piezo2* and *Ntrk1* (the gene that encodes the nerve growth factor receptor TrkA). These results suggest that nerve growth factor-mediated sensitization of joint nociceptors, which is critical for osteoarthritic pain, is also dependent on Piezo2, and targeting Piezo2 may represent a therapeutic option for osteoarthritis pain control.

Introduction: “Nerve growth factor (NGF) has been suggested as a therapeutic target for osteoarthritis pain, and clinical trials with antibodies that neutralize NGF reported positive results in terms of pain relief^{10, 11}. Consistent with this data, several rodent models of joint pain, including models of osteoarthritis pain, have been shown to be dependent on NGF signaling^{12, 13, 14, 15}.”

Discussion first paragraph: “Our observations support a key role for Piezo2 expressed by nociceptors in mediating mechanical sensitization associated with a mouse model of acute inflammatory knee pain, two mouse models of osteoarthritis, as well as with a model induced by local injection of NGF into the knee over 8 weeks. Consistent with this interaction, co-expression of *Piezo2* and *Ntrk1* was demonstrated in subsets of murine as well as human nociceptors. These results support previous work suggesting that Piezo2 is expressed by nociceptors^{4, 20, 24, 25, 52}. However, its primary function on nociceptors does not appear to be the sensation of high intensity mechanical forces under healthy conditions; rather Piezo2 appears to become sensitized in settings of inflammation and tissue damage. This supports previous work

demonstrating that pan-sensory neuron deletion of *Piezo2* reduced mechanical sensitization following acute application of capsaicin and in a model of nerve injury⁴. In addition, individuals with *PIEZO2* loss of function mutations had reduced mechanical allodynia following capsaicin application to the skin³. Osteoarthritis is associated with mechanical sensitization and joint pain with movement. Joint damage products released as a result of ongoing tissue remodeling in osteoarthritis, including NGF, have been implicated in the development of mechanical sensitization^{31, 42}, and NGF signaling has been shown to play an important role in generating osteoarthritis pain⁴³. Our findings suggest that Piezo2 is an essential component of this phenomenon.”

Discussion: “On short time scales, NGF has been shown to trans-activate other channels such as TRPV1. However, chronic sensitization, as assessed in this study through models of experimental osteoarthritis or through repeated intra-articular injections of NGF over the course of 8 weeks, likely involves retrograde transport of NGF to the DRG, where it can promote changes in gene expression and/or membrane localization of channels and upregulation of neuropeptides...Hence, our results suggest that chronic NGF-mediated sensitization of joint nociceptors, which is critical for osteoarthritic pain, is also dependent on Piezo2.”

We have updated the statement in the limitations paragraph to include the levels of NGF reported in the human osteoarthritic joint: “Finally, while the model used here to study the effects of NGF in the knee joint use injections of NGF that are higher than those found in the human osteoarthritis joint (2-6 ng/mL)⁵⁹, this is necessary in order to overcome the initial clearance of the protein⁶⁰ following each injection.”

Reviewer #2 (Remarks to the Author):

The revised manuscript by Obeidat & Wood et al. is fascinating and really adds to the field's knowledge about the role of Piezo2 in mechanical nociception in the context of inflammation. I am impressed by the new experimental models and weight bearing measurement data - they really add interesting and valuable information. I feel that this revised version more strongly supports the author's claims, and has cleared up some of the previously confusing terminology.

My remaining small suggestion is to adjust the data visualization for the calcium imaging in Figure 2. The writing in the heatmap area is quite small. Moreover, the labeling of the start and end of stimuli in different frames is missing from the WT condition.

I am still surprised by the heterozygous results, but the data are clear!

I thank the authors for this nice manuscript and valuable contribution to the field.

Author response and action: Thank you for the positive feedback; we have modified Figure 2 to increase the font size and added labels in the WT condition.